



# Optimizing the Carbonic Anhydrase temperature response and stomatal conductance of carbonyl sulfide leaf uptake in the simple biosphere model (SiB4)

Ara Cho[1], Linda M.J. Kooijmans[1], Kukka-Maaria Kohonen[2], Richard Wehr[3], Maarten C. Krol[1,4]

[1]Meteorology and Air Quality, Wageningen University and Research Centre, Wageningen, The Netherlands
[2]Institute for Atmospheric and Earth System Research/ Physics, Faculty of Science, University of Helsinki, Helsinki, Finland
[3]Center for Atmospheric and Environmental Chemistry, Aerodyne Research, Inc., Billerica, MA, USA
[4]Institute for Marine and Atmospheric Research, Utrecht University, Utrecht, The Netherlands

*Correspondence to*: Ara Cho (ara.cho@wur.nl)

**Abstract.** Carbonyl Sulfide (COS) is a useful tracer to estimate Gross Primary Production (GPP) because it shares part of the uptake pathway with $CO_2$. COS is taken up in plants through hydrolysis, catalyzed by the enzyme carbonic anhydrase (CA), but is not released. The Simple Biosphere model version 4 (SiB4) simulates COS leaf uptake using a conductance approach. SiB4 applies the temperature response of the RuBisCo enzyme (used for photosynthesis) to simulate the COS leaf uptake, but the CA enzyme might respond differently. We introduce a new temperature response function for CA in SiB4, based on enzyme

kinetics with an optimum temperature. Moreover, we determine Ball-Berry model parameters for stomatal conductance ($g_s$) using observation-based estimates of COS flux, GPP, and $g_s$ along with meteorological measurements in an evergreen needleleaf forest (ENF) and deciduous broadleaf forest (DBF). We find that CA has optimum temperatures of 22 °C (ENF) and 38 °C (DBF) with CA's activation energy as 40 kJ mol$^{-1}$, which is lower than that of RuBisCo (45 °C), suggesting that air temperature changes can critically affect CA's catalyzation activity. Optimized values for the Ball-Berry offset parameter $b_0$

(ENF: 0.013, DBF: 0.007 mol m$^{-2}$ s$^{-1}$) are higher (lower) than the original value (0.010 mol m$^{-2}$ s$^{-1}$) in the ENF (DBF), and optimized values for the Ball-Berry slope parameter $b_1$ (ENF: 16.36, DBF: 11.43) are higher than the original value (9.0) at both sites. We apply the optimized $g_{CA}$ and $g_s$ parameters in SiB4 site simulations, thereby improving the timing and peak of COS assimilation. In addition, we show that SiB4 underestimates the leaf humidity stress under conditions where high VPD should limit $g_s$ in the afternoon, thereby overestimating $g_s$. Furthermore, we simulate global COS biosphere fluxes, which show

smaller COS uptake in the tropics and larger COS uptake at higher latitudes, corresponding with the updates made to the CA temperature response. This SiB4 update helps resolve gaps in the COS budget identified in earlier studies. Using our optimization and additional observations of COS uptake over various climate and plant types, we expect further improvements in global COS biosphere flux estimates.



## 1 Introduction

The leaf assimilation of the atmospheric trace gas carbonyl sulfide (COS) has been suggested as a proxy to overcome the limitations of estimating photosynthetic carbon dioxide ($CO_2$) assimilation (Whelan et al., 2018). Observations of the net ecosystem exchange (NEE) of $CO_2$ include both Gross Primary Production (GPP) and ecosystem respiration, and those two individual components cannot be directly observed. COS follows the same diffusional pathway into leaves through plant

stomata as $CO_2$. COS is then destroyed through hydrolysis catalyzed by the enzyme carbonic anhydrase (CA) and is assumed not to be produced by any process within leaves (Protoschill-Krebs et al., 1996; Stimler et al., 2010). The CA chemistry is not light-dependent (Protoschill-Krebs et al., 1996), in contrast to photosynthetic $CO_2$ fixation, which requires light. Therefore, measurements of COS uptake can provide information on stomatal conductance, e.g. during the night (Kooijmans et al., 2017), which cannot be obtained from $CO_2$ measurements.


Atmospheric COS mole fractions vary around 500 parts per trillion (ppt) and are primarily influenced by biosphere uptake, ocean emissions, and anthropogenic emissions (Kettle et al., 2002). Recent studies have found that a source is missing in the tropical region (Berry et al., 2013; Glatthor et al., 2015; Kuai et al., 2015; Ma et al., 2021). Moreover, Berry et al. (2013) showed that a sink is missing, or a source is overestimated at higher latitudes. These findings ask for careful evaluation of all

sources and sinks, including the biosphere.

Biosphere models, such as the Simple Biosphere model, version 4 (SiB4) (Berry et al., 2013; Kooijmans et al., 2021) and the Organizing Carbon and Hydrology In Dynamic Ecosystems model (ORCHIDEE; Launois et al., 2015; Maignan et al., 2021, Remaud et al., 2021, Abadie et al., 2021) have been used to estimate ecosystem exchange of COS quantitatively. The SiB4

COS biosphere exchange was recently assessed against observations by Kooijmans et al. (2021). They stressed the need to account for spatial and temporal variations in atmospheric COS mole fractions, which largely reduce SiB4 COS biosphere uptake in the tropics (although observations to confirm this influence are lacking). The calculated reduction in the tropics was not large enough to explain the gap in the COS budget. Kooijmans et al. (2021) and Vesala et al., (2022) also found that SiB4 COS biosphere flux simulations were low compared to observations in the boreal region, consistent with the underestimations

found by Ma et al. (2021). Our study follows one of the recommendations in Kooijmans et al. (2021) by focusing on the parameterization of the temperature dependence of the CA enzyme activity to improve simulations of the vegetation COS uptake in SiB4.

In SiB4, the COS assimilation is described as a series of resistances (i.e. inverse conductances) at the leaf boundary layer ($g_b$),

the stomatal pores ($g_s$), and the leaves' interior ($g_{CA}$). These conductances to COS are scaled relative to conductances for water vapor or $CO_2$ with diffusivity ratios and a calibration factor. For $g_{CA}$, previous studies found that both the CA enzyme activity (Badger and Price 1994) and mesophyll conductance (Evans et al., 1994) scale with the maximum velocity of carboxylation





($V_{max}$) by the enzyme RuBisCo. Therefore, the COS internal conductance in SiB4 is scaled to $V_{max}$ through a single calibration factor $\alpha$ based on laboratory leaf gas exchange measurements (Stimler et al., 2010, 2011, Berry et al., 2013). However, the

enzymatic control of COS and $CO_2$ assimilation differ. COS molecules are hydrolyzed by the enzyme CA in the mesophyll cells (Protoschill-Kreb et al., 1996). In contrast, photosynthesis is further controlled by the enzyme RuBisCo. Thus, $CO_2$ has a different point of uptake compared to COS. The enzyme activity depends on the enzyme abundance and is related to environmental parameters such as temperature and pH (Michaelis and Menten, 1913). In particular, the CA enzyme does not require light to catalyze COS hydrolysis, whereas the RuBisCo enzyme does require light (Stimler et al., 2010).


Several studies found that the leaf relative uptake ratio (LRU, which is proportional to the ratio of COS and $CO_2$ deposition velocities) varies with temperature under conditions where light was not limiting photosynthesis (Cochavi et al., 2021; Stimler et al., 2010; Sun et al., 2018; Kooijmans et al., 2019). More specifically, the LRU decreased with increasing temperatures above 15 ℃, indicating that COS uptake has a lower optimum temperature than $CO_2$ uptake, possibly driven by different

temperature responses of the CA and RuBisCo enzymes. Therefore, to accurately simulate the relation between COS and $CO_2$ exchange in leaves, it is necessary to use separate temperature response equations for the internal conductance to $CO_2$ and COS.

Besides uncertainties in $g_{CA}$, uncertainties in $g_s$ can also affect the accuracy of simulated COS assimilation. A common

approach for simulating $g_s$ is the semi-empirical Ball-Berry (BB) model (e.g. Ball et al., 1987; Ball 1988; Collatz et al., 1992). This model is also applied in SiB4 and utilizes a set of related variables (e.g. relative humidity, $CO_2$ concentration at the leaf surface, photosynthesis) and two empirical constants. One of the constants ($b_1$) describes the slope of the relation between $g_s$ and GPP. The other constant ($b_0$) represents the residual $g_s$ in the dark. The current implementation of the BB model in SiB4 has only one pair of $b_1$ and $b_0$ values for C3 plants and only one pair for C4 plants, whereas the BB constants should ideally

be prescribed for each plant functional type (PFT) separately to obtain accurate $g_s$ (Miner et al., 2017). To constrain $b_0$ requires information on nighttime $g_s$. However, obtaining $g_s$ estimates from nighttime water vapor flux measurements in the field is highly uncertain due to observational constraints (Papale et al., 2006; Wehr et al., 2017; Wehr and Saleska, 2021). As an alternative, nighttime COS uptake was previously reported (White et al., 2010; Belviso et al., 2013; Commane et al., 2013, 2015; Berkelhammer et al., 2014; Billesbach et al., 2014; Wehr et al., 2017; Kooijmans et al., 2017), and when the soil uptake

is properly accounted for, this flux could provide information on stomatal opening. Several multi-year measurement datasets of $CO_2$ and COS biosphere and soil fluxes are now available (Commane et al., 2015; Wehr et al., 2017; Vesala et al., 2022), making it possible to use COS to provide information on $g_s$ and thereby constrain the BB model parameters.

This research aims to optimize the temperature response of CA and BB model parameters to better estimate COS assimilation

in the SiB4 model. To do so, we will use observed COS leaf fluxes and GPP, plus observation-based $g_s$. The optimization will be based on observations from two PFTs: a boreal evergreen needleleaf forest (ENF) in Hyytiälä, Finland and a temperate





deciduous broadleaf forest (DBF) at Harvard Forest, USA. The optimized parameters will be applied in the global simulation of the SiB4 biosphere model to evaluate the effects on the global COS biosphere sink.

## 2 Methodology

### 2.1 Modelling COS leaf uptake

### 2.1.1 SiB4 biosphere model

The SiB4 model is a land surface model that calculates the COS flux as described in Berry et al. (2013). The main application of the model is to estimate land-atmosphere exchange of carbon, land surface energy, and water budgets (Sellers et al., 1986; Sato et al., 1989). SiB4 has a timestep of 10-minutes and operates on a spatial resolution of 0.5° x 0.5°. Unlike the previous

SiB3 model, which relies on satellite information, version 4 fully simulates the terrestrial carbon cycle using a process-based model (Haynes et al., 2019).

As each vegetation type has different physiological and phenological characteristics, SiB4 simulates photosynthesis in a heterogenic land cover with different plant functional types (PFTs) per site or grid cell, each with separate fractions. These

PFTs consist of nine natural vegetation classes and three specific crop types (maize, soybeans, and winter wheat), plus the separation of C3 and C4 plants in generic cropland and grassland. Besides responses of plant growth to temperature, humidity, radiation, and precipitation, the model accounts for environmental stress factors as a limitation to plant growth: the leaf humidity stress ($F_{LH}$), the root-zone water stress ($F_{RZ}$), and the canopy temperature stress ($F_T$). Several variables (e.g. $V_{max}$ of RuBisCo) are prescribed according to phenological stages: leaf-out, growth, maturity, senescence, and dormant stages. The

leaf-out stage begins when the environmental conditions are suitable for photosynthesis to take place, and the growth stage is determined when the canopy is large enough to support photosynthesis. The maturity starts when the leaf amount is maintained. When plants experience stress and photosynthetic capacity is reduced, it is prescribed as senescence. In the dormant stage, plants do not have leaves in the canopy, or it has unsuitable conditions for photosynthesis (Haynes et al., 2020).

### 2.1.2 Module for COS vegetation uptake in SiB4

SiB4 simulates COS vegetation assimilation as a combination of three resistances ($g_s$, $g_b$, and $g_{CA}$) multiplied by the atmospheric COS mole fraction (Berry et al., 2013):

$$F_{cos} = C_{cos} \left( \frac{1.94}{g_s} + \frac{1.56}{g_b} + \frac{1}{g_{CA}} \right)^{-1} \tag{1}$$

where $F_{cos}$ is the COS vegetation assimilation in the canopy (pmol m$^{-2}$ s$^{-1}$), and $C_{cos}$ is the atmospheric COS mole fraction in the canopy (pmol mol$^{-1}$). Due to its larger size, COS diffusion through the stomata and the laminar boundary are slowed down

by a factor of 1.94 and 1.56, respectively, relative to water vapor (Seibt et al., 2010; Stimler et al., 2010).





The stomatal conductance $g_s$ (mol m$^{-2}$ s$^{-1}$) in SiB4 is calculated by using the BB model. This model relates $g_s$ and GPP as a function of environmental factors with two empirical constants $b_0$ and $b_1$:

$$g_s = b_1 \frac{GPP_{SiB4}}{CO_{2s}} F_{LH} + b_0 \cdot LAI \cdot F_{RZ} \tag{2}$$

where $GPP_{SiB4}$ (mol C m$^{-2}$ s$^{-1}$) is the canopy CO$_2$ assimilation, $CO_{2S}$ (mol C mol air$^{-1}$) is the CO₂ mole fraction at the leaf surface, $F_{LH}$ (-) is the leaf humidity stress factor, LAI is the leaf area index (-), and $F_{RZ}$ is a non-dimensional term that accounts for root-zone water stress. $F_{LH}$ is related to relative humidity at the leaf surface and is calculated as a ratio of the water vapor mixing ratio at the leaf surface to the water vapor mixing ratio in the leaf internal space (Sellers et al. 1992). The value of $F_{LH}$ for ENF has a lower bound of 0.7, making ENF more resilient to humidity stress. However, Smith et al. (2020) found that with

the 0.7 threshold in place, SiB4 did not accurately simulate a drought response for European ENF ecosystems. Therefore, we removed this lower bound in the optimization, but will show the impact in a sensitivity study in Sect. 3.5.1.

The empirical constant $b_1$ is the slope of the linear relationship between $g_s$ and $GPP_{SiB4} F_{LH} CO_{2s}^{-1}$ and $b_0$ (mol m$^{-2}$ s$^{-1}$) is the intercept indicating minimum $g_s$ (Ball et al., 1987; Ball 1988). The choice for $b_1$ significantly impacts simulated transpiration

(Leuning et al., 1998; Lai et al., 2000; Bauerle et al., 2014) and is prescribed in SiB4 as 9.0 for C3 plants and 4.0 for C4 plants. The coefficient $b_0$ is 0.01 mol m$^{-2}$ s$^{-1}$ for most PFTs but 0.04 mol m$^{-2}$ s$^{-1}$ for crops and C4 plants. The prescribed $b_0$ term is converted from the leaf to the canopy scale by multiplying by LAI.

$GPP_{SiB4}$ is explicitly calculated in SiB4 using the carbon pool with three assimilation rates limited by enzyme activity ($w_c$),

light ($w_e$), and carbon compound export ($w_s$) (Haynes et al., 2020). The three rates are calculated by functions $f_{c,e,s}$ described in detail in Sellers et al. (1996) depending on a canopy temperature ($T_{can}$, K):

$$w_c = f_c(V_{max}(T_{can}), pCO_{2i}, pO_2(T_{can}), \gamma^*) \tag{3}$$
$$w_e = f_e(APAR, pCO2_i, \gamma^*) \tag{4}$$
$$w_s = f_s(V_{max}(T_{can}), F_{RZ}, pO_2(T_{can})) \tag{5}$$

Where $pCO_{2i}$ (Pa) is the internal partial pressure of CO₂, $pO_2(T)$ (Pa) is the temperature response of partial pressure of O₂, APAR (mol m$^{-2}$ s$^{-1}$) is the absorbed photosynthetically active radiation, and $\gamma^*$ (Pa) is the CO₂ photo-compensation point. The SiB4 model defines GPP as the minimum of these three limiting rates. Note that $GPP_{SiB4}$ is used in SiB4 to calculate the COS leaf flux via $g_s$, as described in Eq. (2) and evaluated independently from GPP calculated by the BB model ($GPP_{BB}$) which will be introduced in Sect. 2.3.1.


The COS molecules that have diffused into the leaf mesophyll cells are hydrolyzed in a reaction catalyzed by the CA enzyme ($g_{CA}$). SiB4 assumes that the $g_{CA}$ (mol m$^{-2}$ s$^{-1}$) scales with $V_{max}$ of RuBisCo at 298 K ($V_{max, rub}$, mol m$^{-2}$ s$^{-1}$) as follows:

$$g_{CA}(PS) = \alpha \cdot V_{max,rub}(PS) \cdot f(T_{can})_{SiB4} \cdot F_{LC}(PS) \cdot F_{RZ} \cdot \left(\frac{P}{P_{sfc}}\right) \cdot \left(\frac{T_{can}}{T_0}\right) \tag{6}$$





$V_{max, rub}$ varies with phenological stage (PS) (see Table 3) and is scaled with a $T_{can}$ response function $f(T_{can})_{SiB4}$ that prescribes

the relative increase per 10 K increase ($Q_{10}$) as 2.1:

$$f(T_{can})_{SiB4} = 2.1^{0.1(T_{can}-298)}$$ (7)

In SiB4, the canopy temperature $T_{can}$ is calculated from the temperature above the canopy, which is normally obtained from a meteorological analysis dataset (Haynes et al., 2020). In this study, however, we use the temperature measured above the canopy to obtain $T_{can}$ by SiB4. Likewise, we use the specific humidity measured above the canopy, which is used by SiB4 to

calculate the leaf humidity stress factor $F_{LH}$ at leaf surface level, needed in Eq. (2).

Other modifying factors in Eq. (6) are the ratio of atmosphere pressure ($P$, hPa) to the surface pressure ($P_{sfc}$ = 1,000 hPa) and the ratio of the temperature to the reference temperature ($T_0$ = 273.15 K). $F_{LC}$ (-) is the scaling factor from leaf to canopy, accounting for a fraction of absorbed PAR (FPAR) and other factors such as light scattering and leaf projection. The calibration

parameter $\alpha$ (-) was obtained from simultaneous measurements of COS and $CO_2$ uptake (Stimler et al., 2010, 2012; Berry et al., 2013) and was estimated as 1400 for C3 and 8862 for C4 plants. These numbers were derived from a limited number of observations, so the values of $\alpha$ do not capture variability between plant species and seasons. Kooijmans et al. (2021) derived $\alpha$ from ecosystem observations of six sites throughout the growing season and found an average $\alpha$ of 1616 ± 562 (C3 plants). Here, the standard deviation indicates large variability over time and between sites. The impact of $\alpha$ on $g_{CA}$ will be described

in Sect. 2.1.3.

**2.1.3 A new approach to describe $g_{CA}$**

Each enzyme has its own kinetic characteristics, with activity generally increasing with temperature up to an optimum temperature and decreasing above this temperature. To derive a more realistic enzyme activity that also accounts for an optimum temperature, we propose a temperature response ($f(T_{can})_{new}$) based on an equilibrium model that applies Michaelis-

Menten kinetics. A similar equilibrium model was previously used in COS soil models (Sun et al., 2015; Ogée et al., 2016) and is described as (Peterson et al. 2004; Daniel et al., 2010):

$$f(T_{can})_{new} = A_T \cdot \frac{T_{can}\exp\left(-\frac{\Delta H_a}{RT_{can}}\right)}{1+\exp\left[-\frac{\Delta H_{eq}}{R}\left(\frac{1}{T_{can}}-\frac{1}{T_{eq}}\right)\right]}$$ (8)

Here, three variables for enzyme kinetics are included: $\Delta H_a$ (kJ mol⁻¹) is the activation free energy of the CA enzyme; $\Delta H_{eq}$ (kJ mol⁻¹) is the enthalpy change when the enzyme converts from an activated to inactivated state; $T_{eq}$ (K) is the temperature

at which activated and inactive enzymes' concentrations are equal (Daniel et al., 2010; Sun et al., 2015). The factor $A_T$ normalizes Eq. (8) such that, equivalent to Eq. (7), $f(T_{can})_{new}$=1 at $T_{can}$ = 298 K. We adopt $A_T$ as the value of $f(T_{can})_{new}$⁻¹ when $T$ is equal to $T_{eq}$. $R$ is the universal gas constant (8.3145 J K⁻¹ mol⁻¹). Figure 1 shows that $\alpha$ and the three kinetic parameters have different effects on the temperature response of $g_{CA}$ (Eq. 6). The calibration parameter $\alpha$ affects the strength of $g_{CA}$ (Fig. 1(a)) and its accuracy is therefore crucial for accurate COS flux simulations. With $\Delta H_a$ increasing (Fig. 1(b)), $g_{CA}$ decreases





(increases) for temperatures above (below) the optimal temperature. $\Delta H_{eq}$ has the opposite effect, albeit with a different response to $\Delta H_a$ (Fig. 1(c)). Both $\Delta H_a$ and $\Delta H_{eq}$ affect $g_{CA}$ depending on the temperature range. Finally, Fig. 1(d) shows that $T_{eq}$ determines the optimum of the temperature response curve without having impact on the magnitude of $g_{CA}$.

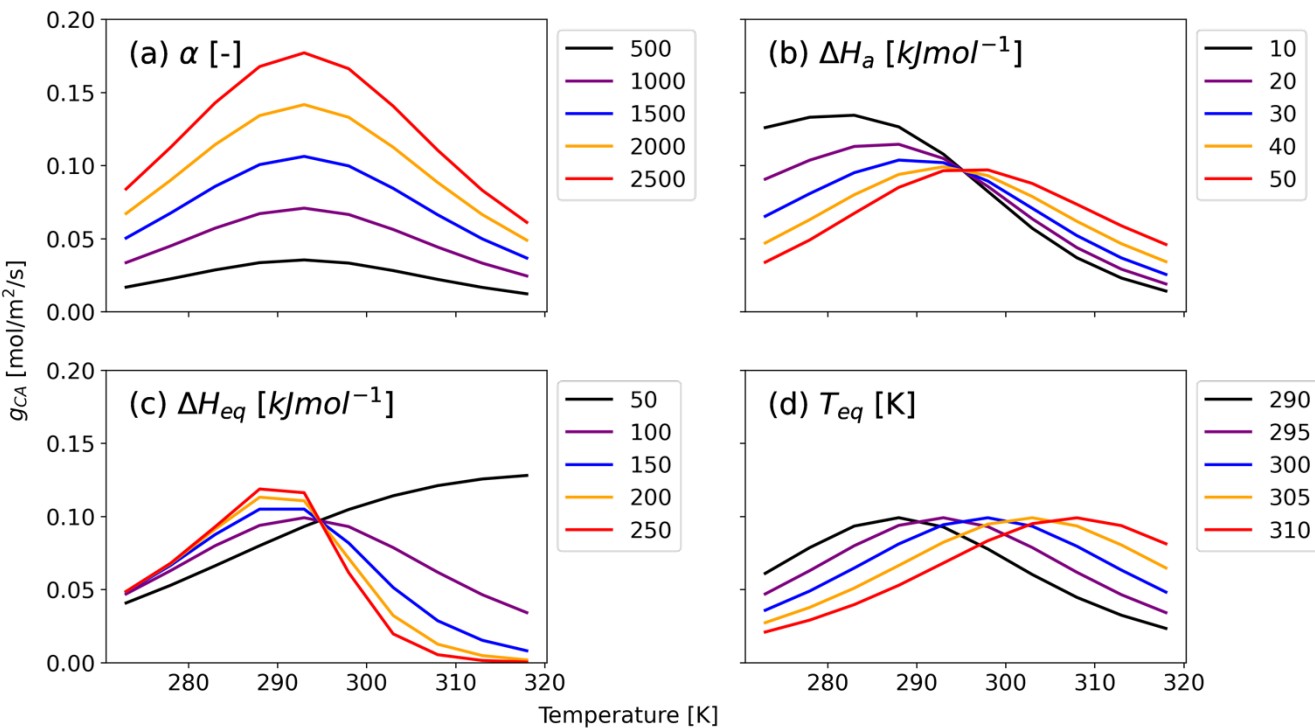

**Figure 1: Calculated $g_{CA}$ as a function of canopy temperature for parameters (a) $\alpha$, (b) $\Delta H_a$, (c) $\Delta H_{eq}$, and (d) $T_{eq}$ from Eq. (6), (8). Each parameter is set to five different values given in the caption to investigate the response of $g_{CA}$ to temperature. While the target parameter changes, the other variables are fixed as $\alpha = 1400$ $\Delta H_a = 40$ kJ mol$^{-1}$ $\Delta H_{eq} = 100$ kJ mol$^{-1}$, and $T_{eq} = 295$ K which are initial values for Hyytiälä (see Sect. 2.3.2).**

## 2.2 Observation

### 2.2.1 Observed variables

In the optimization of $g_s$ and $g_{CA}$, we used observed values of the variables required in the COS leaf uptake calculation (Eq. (1)), namely: COS ecosystem flux, COS soil flux, GPP, $C_{cos}$, temperature, and specific humidity. The observations were obtained at Hyytiälä in Finland during 2013-2017 (Kooijmans et al., 2017; Sun et al., 2018; Vesala et al., 2022) and at the Harvard Forest in the United States during 2012 and 2013 (Commane et al., 2015; Commane et al., 2016; Wehr et al., 2017). COS and GPP ecosystem fluxes were measured with the eddy-covariance (EC) technique.



For Hyytiälä, the EC processing steps were described by Kohonen et al. (2020) and Vesala et al. (2022) and GPP was derived from NEE using multi-year parameter fits (Kolari et al., 2014, Kohonen et al., 2022). The effect of storage in the canopy airspace was corrected by collocated COS profiles (Kooijmans et al., 2017; Kohonen et al., 2020).

210

For the Harvard Forest, we used GPP derived from $CO_2$ isotope EC measurements as reported in Wehr et al. (2016), and we used canopy COS uptake derived from COS EC measurements as reported in Wehr et al. (2017).

In addition to the COS ecosystem fluxes, COS soil flux measurements were available for the 2016 growing season at Hyytiälä, and for the 2012 and 2013 growing seasons the Harvard Forest.

215    To ensure data quality for the COS ecosystem flux, soil flux, GPP, and COS mole fraction, we used three-hourly averages each month for each observed variable. We removed outliers that fell out of the 25 to 75 percentile range in each three-hourly period. We only used data points when more than three data points were present at three-hourly time intervals in each month and when all variables required for the optimization were available.

220    Figure 2 shows the resulting average diurnal cycle per month for COS ecosystem, soil, and vegetation fluxes (ecosystem flux minus soil flux). Note that positive fluxes indicate uptake. As the seasonal and diurnal variations in COS soil fluxes were small (Sun et al., 2018), we applied the monthly average diurnal cycle of the soil flux from 2016 to the other years (2013-2015 and 2017).





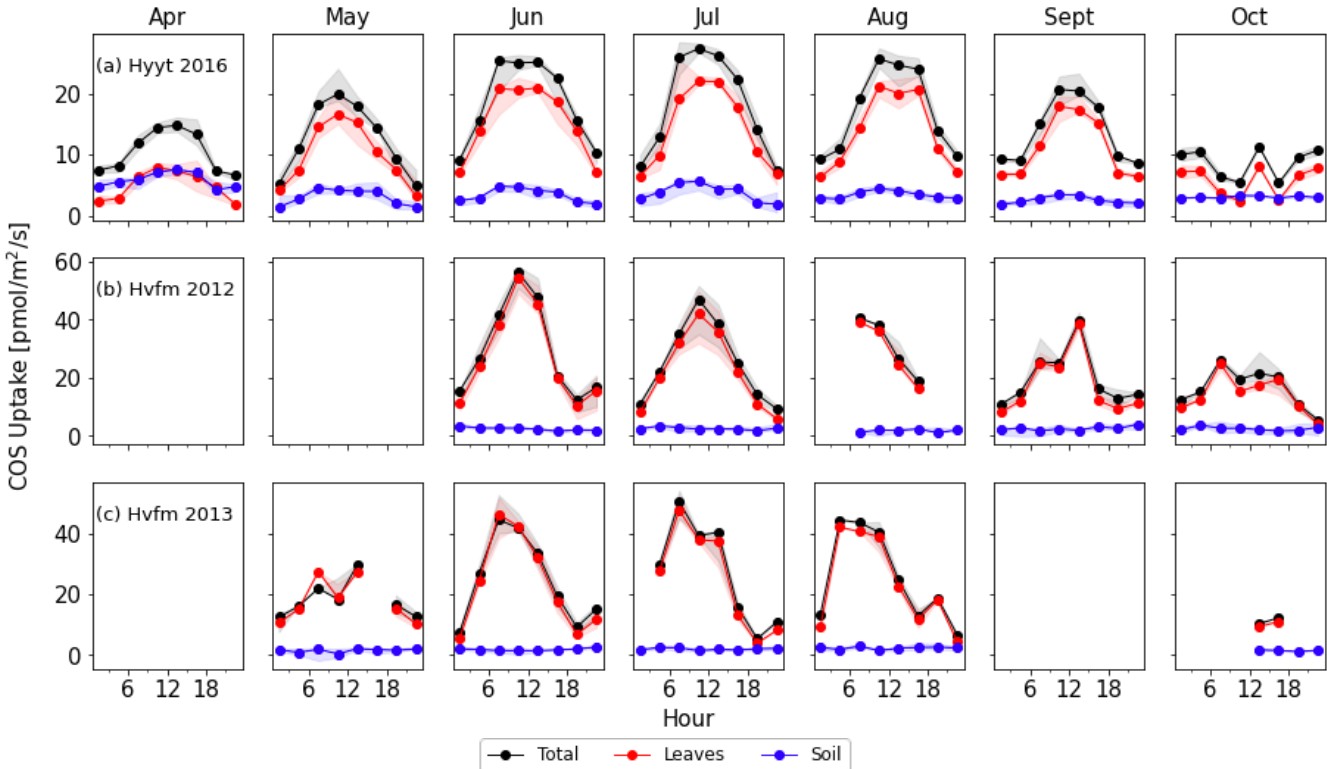

**Figure 2. Monthly diurnal variation of COS fluxes in 2016 at Hyytiälä (a) and 2012-2013 at Harvard Forest (b, c). Lines are median values, and filled areas show the 25 to 75 percentile range of the data. Black: COS ecosystem flux, blue: soil flux, red: vegetation flux estimated as ecosystem minus soil flux.**

### 2.2.2 Observation-based $g_s$ and $g_{CA}$

Observation-based $g_s$ was derived from sensible heat flux and evapotranspiration measurements using the Flux Gradient (FG) equations (Baldocchi et al., 1991; Wehr and Saleska, 2015; Wehr and Saleska, 2021). A key step in the derivation of $g_s$ is the estimation of transpiration from evapotranspiration. At the Harvard Forest, transpiration was estimated by an empirical equation established during times of minimal non-stomatal evaporation (i.e. a few days after rain, removing mornings with dew evaporation), as described in Wehr et al. (2017). At Hyytiälä, we simply restricted our analysis to periods of minimal non-stomatal evaporation by eliminating data when the dew point was equal to or greater than the air temperature or when the accumulated precipitation for the past two days was more than 0.01 mm.

The FG approach leads to significant uncertainties for nighttime data because the leaf to air-water vapor gradient is too small under stable conditions (Wehr et al., 2017). We thus excluded nighttime $g_s$ when the values were smaller than 0.05 mol m$^{-2}$ s$^{-}$





[1]. To reduce the effect of random noise on $g_s$, we used an average diurnal cycle (based on three-hourly medians) for each month.

Observation-based $g_{CA}$ was extracted by rewriting Eq. (1):

$$g_{CA} = \left( \frac{C_{cos}}{F_{cos}} - \frac{1.94}{g_s} - \frac{1.56}{g_b} \right)^{-1} \qquad (9)$$

Here we used the observation-based $g_s$ from the FG equation as discussed above and filtered observations of $C_{cos}$ and $F_{cos}$. Additionally, we used simulated $g_b$ from SiB4, as we don't have observed $g_b$ available and as the value of $g_b$ only has a minor effect on $F_{cos}$, which will be further discussed in Sect. 3.1. Although outliers of observed $g_s$, $C_{cos}$, and $F_{cos}$ were removed

already, a significant number of outliers in $g_{CA}$ appeared because of error propagation. We removed additional outliers outside the 25–75 percentile range of the $g_{CA}$ dataset.

**2.3 Optimization**

**2.3.1 Procedure**

In the optimization steps, we minimized a quadratic cost function $J(x)$ based on Bayes' theorem (Tarantola and Vallette 1982,

Enting et al., 1993):

$$J(x) = \frac{(x - x_a)^2}{2\sigma_a^2} + \frac{(y - H(x))^2}{2\sigma_y^2} \qquad (10)$$

Here, $x$ represents the state, $x_a$ the prior settings of the state, and $\sigma_a$ the error assigned to the parameters. In the second term, $y$ represents the observations and $H(x)$ the model evaluation using the state $x$. The error $\sigma_y$ represents the observational error. The details of $\sigma_a$ and $\sigma_y$ will be described in Sect. 2.3.2 and Sect. 2.3.3, respectively.


To optimize the $g_s$ and $g_{CA}$ parameters, we intend to use the information from GPP and COS leaf uptake measurements simultaneously. Thus, we propose a two-step approach in combination with an iterative minimization of the cost functions, as outlined in Fig. 3. In the first step, we optimally estimate $g_s$ parameter $b_1$ by minimizing $J(x)$ which sums GPP differences between estimation ($H(b_1)$ in Eq. (10)) and observation. We select to use GPP observations in the optimization over $g_s$ because

GPP can be used to evaluate $b_1$ and $b_0$ using the BB model. Moreover, observed GPP leads to more accurate $b_1$ values, because of uncertainties in GPP are smaller compared to observation-based $g_s$. Here, we use only positive $GPP_{obs}$ values (uptake) because the BB model is only applicable in daytime conditions. The estimated GPP ($H(b_1)$ in Eq. (10)) is calculated by rewriting the BB model using observation-based $g_s$ (Sect. 2.2.2), modelled RH at the leaf surface ($F_{LH}$), and simulated $CO_{2S}$ from SiB4. Hereinafter the estimated GPP by the BB model is called $GPP_{BB}$ :

$$GPP_{BB} = \frac{(g_s - b_0 \cdot LAI \cdot F_{RZ}) \, CO_{2S}}{b_1 \cdot F_{LH}} \qquad (11)$$



In the second optimization step, we optimize the $b_0$ and $g_{CA}$ parameters ($\alpha$ in Eq. (6) and $T_{eq}$ in Eq. (8)). These parameters are optimized by minimizing the differences between calculated and observed $F_{COS}$. $F_{COS}$ is calculated with three conductances using Eq. (1). Specifically, $g_s$ is estimated with Eq. (2) using the optimized $b_1$ from step 1. Here, we used $GPP_{SiB4}$ to satisfy

our aim of optimizing the SiB4 model parameters. Note that $GPP_{BB}$ from Eq. (11) cannot be used here, because it would make the estimated $g_s$ equal to observation-based $g_s$. Based on sensitivity studies in Appendix A, we decided to select $\alpha$, $b_0$, and $T_{eq}$ as target parameters and to fix $\Delta H_{eq}$ and $\Delta H_a$ at 100 kJ mol$^{-1}$ and 40 kJ mol$^{-1}$, respectively.

In the optimization procedure, we specifically exploit the fact that the nighttime COS flux carries information about nighttime

$g_s$ through the parameter $b_0$. The alternative, i.e. optimizing $b_0$ already in step 1, would ignore the information of nighttime $g_s$ brought by COS flux observations. Consequently, however, we have to iterate the procedure several times to reach convergence. Figure 3 specifies which observations are used in which step ($g_s$, $GPP_{obs}$, $F_{COS, obs}$, $C_{cos}$ highlighted as grey) and which variables are simulated by SiB4 (e.g. $T_{can}$, $F_{LH}$, $GPP_{SiB4}$, $CO_{2S}$, $g_b$).

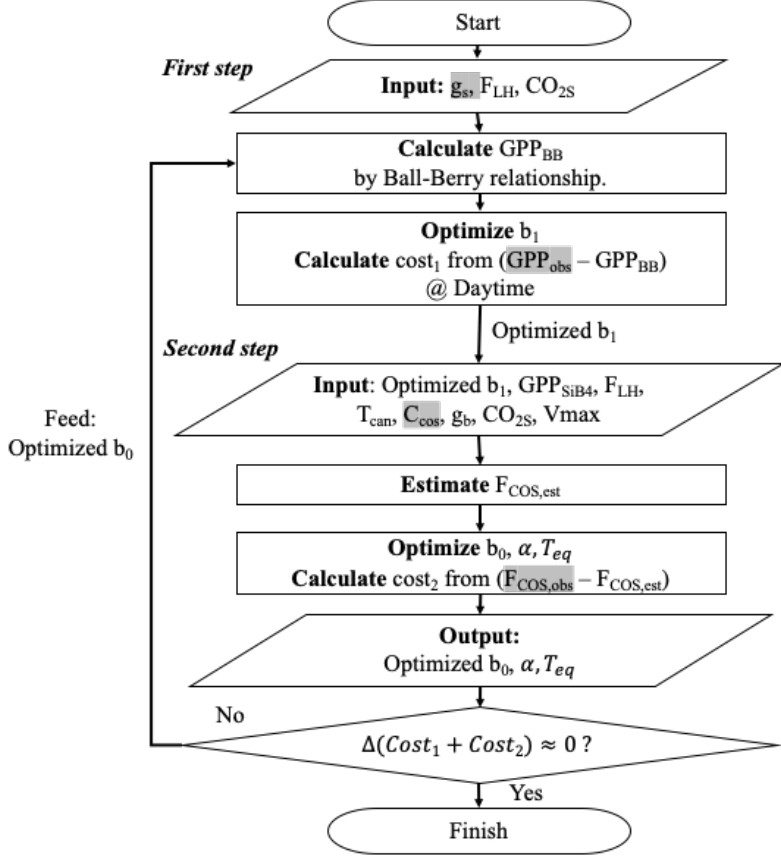

**Figure 3. Flow chart of the procedure to optimize COS leaf uptake's parameters. The procedure has two steps: (1) Optimize $b_1$ by minimizing deviations between GPP$_{BB}$ and observations, (2) Optimize $b_0$, $\alpha$, and $T_{eq}$ by minimizing deviations between modelled and observed COS uptake. Variables highlighted in grey are from observations, and the other variables are estimated from SiB4.**





We applied the Simplicial homology global optimization (SHGO) from the SciPy python library to minimize the cost functions.
SHGO is appropriate for solving non-continuous, non-convex and non-smooth functions (Endres et al., 2018). SHGO also allows the definition of a valid parameter range, as will be discussed in Sect. 2.3.2 and in Appendix A.

The $V_{max}$ of RuBisCo was found to vary over the phenological stage and per PFT (Woodward et al., 1995; Wolf et al., 2006; Kattge et al., 2009; Walker et al., 2014), which also affects the calibration factor $\alpha$. Therefore, we optimized $\alpha$ for each PFT
and each phenological stage. In contrast, $b_0$, $b_1$, and $T_{eq}$ were only separately determined for the different PFTs, assuming local characteristics for each PFT.

### 2.3.2 Initial parameters and prior errors

The first term in the cost function (Eq. (10)) ties the values of the parameters to realistic values. We additionally confined the parameter values within realistic physical ranges using the SHGO algorithm. Initial parameters and prior errors were chosen
based on thresholds outlined in Appendix A, and they will be compared with optimized results in Sect. 3.3. The variation in the resulting cost function shows distinct differences between Hyytiälä and Harvard Forest, which reinforces our strategy to optimize parameters for each station separately.

### 2.3.3 Observation Errors

To quantify the observational errors $\sigma_y$, we first calculated the three-hourly average coefficient of variation (CV) relative to
the mean of the observed COS vegetation flux in each phenological stage and observed GPP for the entire growing season. Figure 4 shows the results of observational errors. The GPP error is applied in step 1 and the COS leaf uptake error is used in step 2 in the optimization. We multiplied the CV with the mean in each phenological stage. Here, we classify the error of the COS leaf uptake in each phenological stage because we optimized $\alpha$ in each stage. In Fig. 4, we found that the errors differ slightly per phenological stage. In Hyytiälä, the errors are larger in the growth stage compared to the maturity stage, possibly
due to the unstable weather conditions in growth stage. The COS leaf uptake error is larger at both stations during nighttime than during daytime. A potential reason can be the relatively higher uncertainty in the eddy covariance method during stable nighttime conditions.





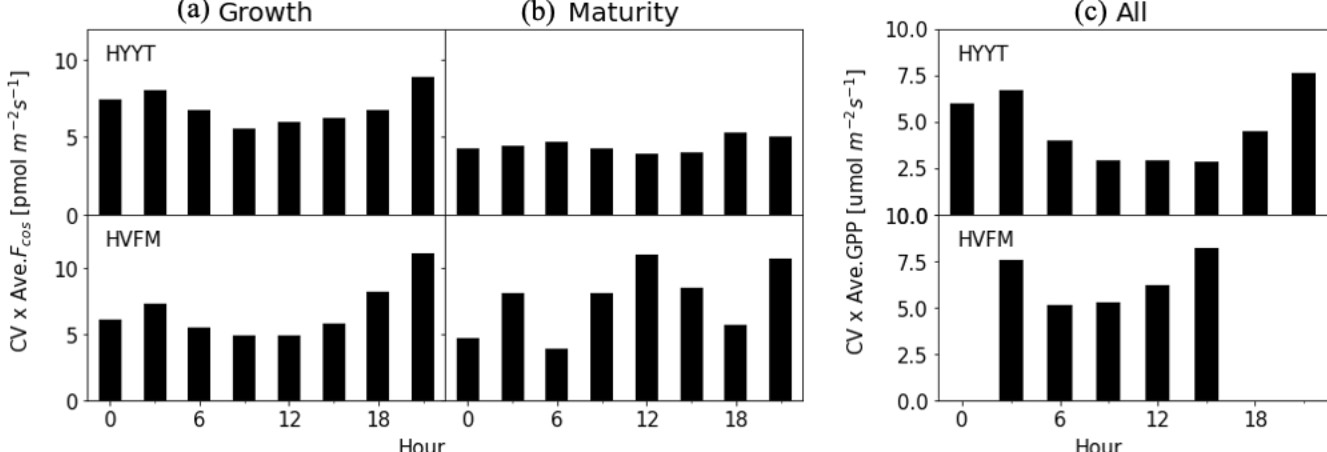

**Figure 4. Distribution of the observation error in the growth (a) and maturity stages (b) for COS leaf uptake and all stages for GPP (c). The upper panel is the errors for Hyytiälä (HYYT) and the lower panel is for Harvard Forest (HVFM).**

## 2.4 SiB4 simulations

We utilized several simulated variables from SiB4 in our optimization. Specifically, calculated $GPP_{SiB4}$, $g_b$, $T_{can}$, $V_{max}$ of RuBisCo, functions $F_{LH}$, $F_{RZ}$ and $F_{LC}$ were used to calculate COS leaf uptake. In addition, LAI and $CO_{2S}$ were used to estimate $GPP_{BB}$. Furthermore, we introduced the new temperature function ($f(T_{can})_{new}$) in the $g_{CA}$ calculation (Sect. 2.2.3) to calculate COS leaf uptake, and excluded $P\,P_{sfc}^{-1}$ and $T_{can}\,T_0^{-1}$ from Eq. (3) due to minor impacts of these factors for these ecosystems.

To simulate the SiB4 data at the two stations, we used the Modern-Era Retrospective Analysis for Research and Application, version 2 (MERRA-2) (Gelaro et al., 2017) as meteorological driver data. Only air temperature and leaf specific humidity were taken from observations. To initialize the carbon pools, we spun up the model to equilibrate the pools. The spin-up was performed from 2000 to 2010 with ten iterations. Ambient $CO_2$ mole fractions were prescribed at 370 ppm.

To estimate the global impact of our findings, we performed a global simulation to evaluate COS leaf uptake estimated by the updated $g_s$ and $g_{CA}$ values. The atmospheric COS mixing ratio $C_{cos}$ were taken from optimizations using the TM5 chemical transport model (Ma et al., 2021; Kooijmans et al., 2021). As we found that all target parameters differ per PFT in Sect. 2.3.2, we applied these parameters only to ENF and DBF. However, to confirm the $f(T_{can})_{new}$ effect on COS leaf uptake, we applied $f(T_{can})_{new}$ to all PFTs with averaged optimum $T_{eq}$ from the two stations (303 K) and fixed $\Delta H_{eq}$ (100 kJ mol$^{-1}$) and $\Delta H_a$ (40 kJ mol$^{-1}$) as described in Appendix A. The soil flux is estimated following Ogée et al. (2016) as implemented by Kooijmans et al. (2021).





To examine the humidity impact in SiB4, we performed a simulation with and without the lower threshold for $F_{LH}$ of 0.7 for ENF (see Sect. 2.1.2). Additionally, we replaced the RH at leaf level calculated by SiB4 by RH measured above the canopy.

Results will be shown in Sect. 3.5.1. we simulated the global COS leaf uptake without the 0.7 threshold of $F_{LH}$ for ENF.

## 2.5 Error reduction and statistics

To determine the uncertainty in the optimized model parameters, we employed a Monte Carlo optimization procedure as described in detail in Appendix B. In short, 100 optimizations were performed. In each optimization, we perturbed the state

with random Gaussian noise on the state and the observations (Chevallier et al., 2007; Bosman and Krol, 2022), according to the errors in the state and observations (Fig. 4). Posterior error statistics will be reported in Table 3.

Additionally, we quantified the performance of the optimization by calculating the root mean square errors (RMSEs), mean bias errors (MBEs), and the chi-square metric ($\chi^2$). The $\chi^2$ metric quantifies the average deviation from the observations,

expressed in $\sigma_y$ units. Thus, $\chi^2=1$ signal that, on average, the model fits the observation within $1\sigma$ indicating a realistic error setting.

## 3. Results and discussion

### 3.1 Impact of each conductance

Figure 5 investigates which conductance contributes most to the total conductance ($g_t$). In these plots, all conductances are prior values before optimization. $g_s$ and $g_{CA}$ were derived from observations (Sect. 2.2.2). We find that $g_t$ is determined mainly by $g_{CA}$ and $g_s$. During daytime, $g_{CA}$ is the lowest conductance in almost all months in Hyytiälä but is comparable to $g_s$ in Harvard Forest. The value of $g_b$ is the highest and hence has the smallest impact on $g_t$. This finding supports this study's proposed two-step optimization process to improve $g_s$ and $g_{CA}$ (see Sect. 2.3).






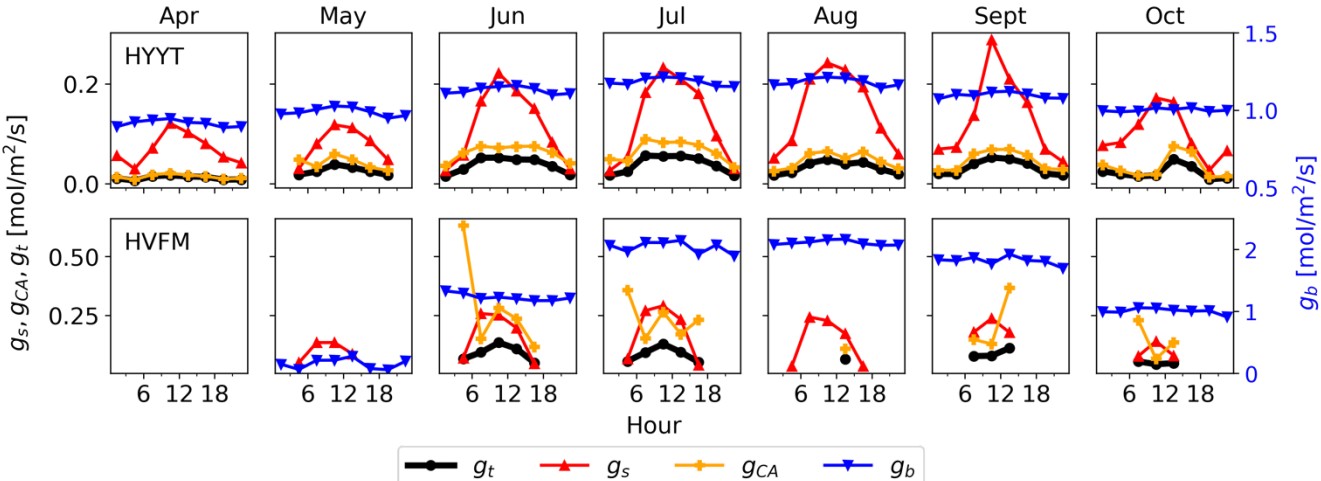

**Figure 5. Monthly averaged diurnal variation of conductances (black: $g_t$, red: $g_s$ orange: $g_{CA}$, blue: $g_b$,) in Hyytiälä (HYYT) and Harvard Forest (HVFM). $g_b$ is estimated by SiB4, $g_s$ and $g_{CA}$ are calculated based on observations as described in Sect. 2.2.2. Negative values are not displayed. The total conductance $g_t$ is calculated from $g_b$, $g_s$, and $g_{CA}$ according to Eq. (1).**

**3.2 Optimization performance**

We obtained optimized parameters after five iterations. By design, the optimized results reduced the deviations between model and observation of GPP and COS leaf uptake. This improvement is quantified by statistical indexes in Tables 1 and 2, respectively. $GPP_{BB}$ is improved compared to the prior (Table 1), with a slight RMSE reduction from 5.51 to 4.87 $\mu mol\ m^{-2}\ s^{-1}$ and a MBE reduction from 0.69 to 0.04 $\mu mol\ m^{-2}\ s^{-1}$ (Table 1). The $\chi^2$ was reduced from 0.88 to 0.80. The

improvement in $GPP_{BB}$ reflects the effect of optimizing $b_1$ and $b_0$ in the BB model.

**Table 1. RMSE, MBE, and $\chi^2$ for the estimation of $GPP_{BB}$ in daytime using prior stomata parameters (Pri) and posterior parameters (Post).**

| Type | RMSE ($\mu mol\ m^{-2}\ s^{-1}$) | MBE ($\mu mol\ m^{-2}\ s^{-1}$) | $\chi^2$ |
|---|---|---|---|
| Prior (Pri) | 5.51 | 0.69 | 0.88 |
| **Posterior (Post)** | **4.87** | **0.04** | **0.80** |

The posterior result of COS leaf uptake ("Post" in Table 2) shows a slight improvement compared to the original state variables with $f(T_{can})_{new}$ in RMSE (from 8.23 to 6.43 $pmol\ m^{-2}\ s^{-1}$) but significantly improved MBE (from -4.65 to -0.34 $pmol\ m^{-2}\ s^{-1}$, see "Post" in Table 2). The large RMSE reflects the typically large random noise of COS flux observations (Kooijmans et al., 2016; Kohonen et al., 2020). However, $\chi^2$ drops from 1.01 to 0.61, confirming that the optimization properly reduced the mismatch between observations and the model within the error statistics. Figure 6 compares the optimized COS

leaf uptake to the original SiB4 simulation in scatter plots. Where the original simulation with $f(T_{can})_{SiB4}$ and previous state





variables was often underestimating the observations, the optimized results resemble the observations over a larger range of the data.

**Table 2. Same as Table 1 but for COS leaf uptake, as applied in the original f($T_{can}$)$_{SiB4}$ simulation with original conductance**
**parameters (Org), with the new temperature response function $f(T_{can})_{new}$ using the initial $g_s$ and $g_{CA}$ parameters (Pri), with optimized parameters (Post). Here, state parameters relevant for $g_{CA}$ are $\alpha$ and $T_{eq}$. The state parameters relevant for $g_s$ are $b_0$ and $b_1$.**

| Type | $g_{CA}$ f(T) | RMSE ($pmol\ m^{-2}\ s^{-1}$) | MBE ($pmol\ m^{-2}\ s^{-1}$) | $\chi^2$ |
|---|---|---|---|---|
| Previous SiB4 (Org) | $f(T)_{SiB4}$ | 8.23 | -4.65 | 1.01 |
| Prior (Pri) | $f(T)_{New}$ | 6.71 | 1.32 | 0.66 |
| Posterior (Post) | | 6.43 | -0.34 | 0.61 |

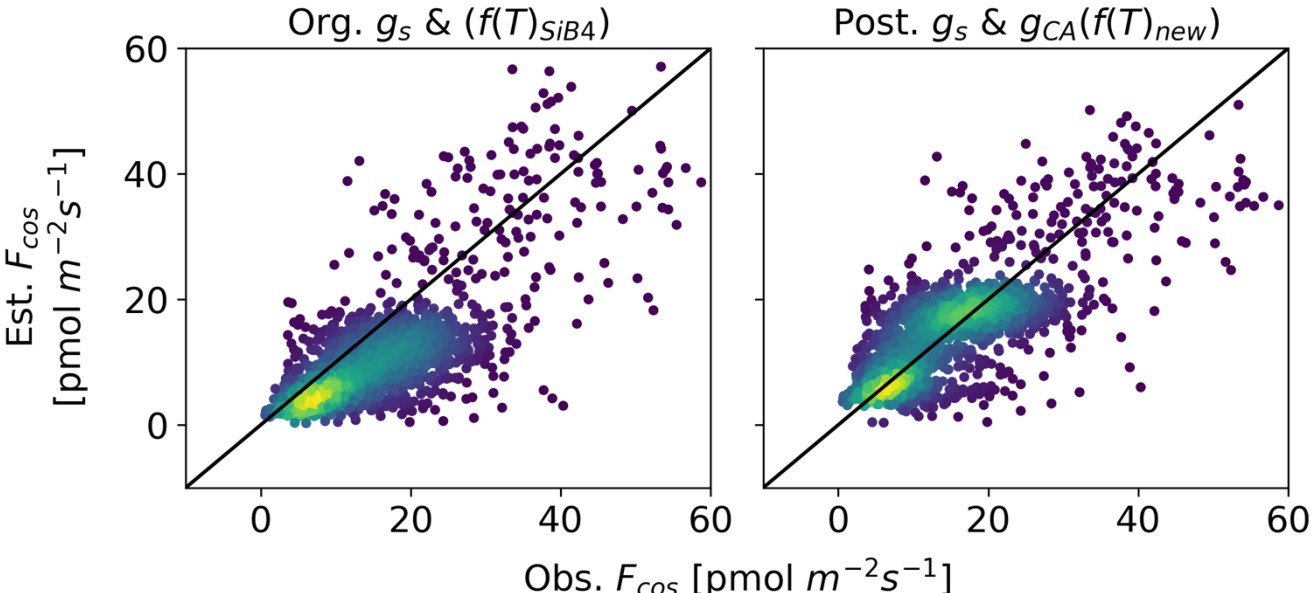

**Figure 6. Scatter plots between observed and estimated COS leaf uptake from original parameters with $f(T_{can})_{SiB4}$ (left) and optimized parameters with $f(T_{can})_{new}$ (right). The colors represent the density of data.**

### 3.3 Optimized parameters

The optimized parameter values with posterior errors are listed in Table 3. The optimized SiB4 parameters differ between the
stations, likely because the dominant PFT and the climate conditions differ between Hyytiälä and Harvard Forest. For instance, the optimized $T_{eq}$ is smaller in Hyytiälä (295 K) than in Harvard Forest (311 K). Thus, the optimum temperature reflects temperature dependence of the enzyme and its adaptation to temperature (Lee et al., 2007). This indicates that regional climate information is important for the correct estimation of $g_{CA}$. The range of $T_{eq}$ can be compared with other COS soil models. For



instance, Ogée et al. (2016) used the same prescribed value for $\Delta H_a$ and they adopt $T_{eq}$ as 298 K, which is in the middle of our
optimized temperatures for Hyytiälä and Harvard Forest.

The $\alpha$, which is the enzyme activity of CA relative to the $V_{max}$ of RuBisCo, is reduced from 1400 to 1316 (in growth) and 1331 (in maturity) in Hyytiälä. In Harvard Forest, $\alpha$ values are larger than the original values in SiB4 for leaf-out (1798), growth (1740), and maturity (2224) phenological stages. Here it should be noted that the change of $\alpha$ should be interpreted in
combination with the new temperature function $f(T_{can})_{new}$ of $g_{CA}$. Since we only optimize $T_{eq}$ for two PFTs (with identical and fixed values for $\Delta H_a$ and $\Delta H_{eq}$) and $T_{eq}$ only shifts $f(T_{can})_{new}$ (Fig. 1), the magnitude of $g_{CA}$ is primarily determined by parameter $\alpha$. Since the observed COS leaf uptake in Harvard Forest is larger than in Hyytiälä, larger values of $\alpha$ are derived for Harvard Forest. The different values of $\alpha$ derived for different phenological stages will be discussed in Sect. 3.4.

The optimized results of the BB model parameters $b_0$ are similar but $b_1$ values are mostly higher than the original values used in SiB4. The parameter $b_0$ for Hyytiälä (0.013 mol m$^{-2}$ s$^{-1}$) and Harvard Forest (0.007 mol m$^{-2}$ s$^{-1}$) are higher and smaller compared to the initial value (0.010 mol m$^{-2}$ s$^{-1}$). For the optimized BB model parameter $b_1$, the empirical slope between $g_s$ and GPP, we find a considerable increase in Hyytiälä (16.38) and a slight increase in Harvard Forest (11.43), compared to the prescribed SIB4 value of 9.0. Our optimized values are larger than the values presented in a review paper for the evergreen
gymnosperm tree which showed $b_1 = 6.8$ and are similar to $b_1 = 8.7$ for the deciduous angiosperm tree (Miner et al., 2017). As will be discussed in Sect. 3.5, the higher slope in Hyytiälä is possibly related to an incomplete separation of observed transpiration rates from the latent heat flux.

Concerning the estimated errors in $b_0$, $b_1$, and $T_{eq}$, we find that errors have been reduced significantly compared to the prior
error range. This indicates that the available data constrain these parameters well. Only the $\alpha$ parameters of Harvard forest are less well constrained. Also, the skill of the optimization to independently optimize the parameters is high, as quantified by the posterior covariances that are presented in Appendix B.

**Table 3. Original (Org) and optimized (Post) state vectors for Hyytiälä and Harvard Forest in different phenological stages as defined**
**by SiB4. Values of Posterior in parenthesis indicates posteriori errors. Detailed error reduction is described in Appendix B.**

| Approach | State vector | Hyytiälä | | Harvard Forest | | |
|---|---|---|---|---|---|---|
| | | Growth | Maturity | Leaf-out | Growth | Maturity |
| Previous SiB4 (Org) | $V_{max}$ of RuBisCo ($\mu$mol m$^{-2}$ s$^{-1}$) | 52 | 54 | 96 | 94 | 92 |
| | $\alpha$ (-) | | | 1400 | | |
| | $b_0$ (mol m$^{-2}$ s$^{-1}$) | | | 0.01 | | |
| | $b_1$ (-) | | | 9.0 | | |
| Prior | $\alpha$ (-) | 1400 ($\pm$ 1000) | | 2000 ($\pm$ 1000) | | |





| (Pri) | $b_0$ (mol m$^{-2}$ s$^{-1}$) | 0.02 (± 0.02) | | 0.01 (± 0.02) | | |
|---|---|---|---|---|---|---|
| | $b_1$ (-) | 17 (± 5) | | 12 (± 4) | | |
| | $T_{eq}$ (K) | 295 (± 20) | | 310 (± 20) | | |
| Posterior (Post) | $\alpha$ (-) | 1316 (± 509) | 1331 (± 574) | 1798 (± 527) | 1740 (± 494) | 2224 (± 613) |
| | $b_0$ (mol m$^{-2}$ s$^{-1}$) | 0.013 (± 0.009) | | 0.007 (± 0.006) | | |
| | $b_1$ (-) | 16.36 (± 2.87) | | 11.43 (± 1.98) | | |
| | $T_{eq}$ (K) | 295 (± 11) | | 311 (± 10) | | |

### 3.4 Optimized temperature response

The optimized parameters show significant improvement in temperature response of the COS leaf uptake. Figure 7 presents the temperature dependency of $g_{CA}$ and COS leaf uptake from the original and optimized simulations output and observations.

The observed COS leaf uptake and pseudo-observations $g_{CA}$ (details in Sect. 2.2.2) in Hyytiälä show a decrease above 20 ℃. As stated before, the original $f(T_{can})_{SiB4}$ describes the CA enzyme activity as an exponentially increasing response to temperature, which does not resemble the observations. The optimized $g_{CA}$ and COS leaf uptake follow the temperature dependence of the observation more closely than the original $f(T_{can})_{SiB4}$. In Harvard Forest, an underestimated bias is shown at a lower temperature under 10 ℃, mostly corresponding to nighttime. This underestimate is related to the uncertainty in

nighttime $g_S$ and the small data volume at low temperatures (details in Sect. 2.2.2).

In the upper panel of Fig. 7, we see the different roles of $\alpha$ and $f(T_{can})_{new}$ in the improvement of $g_{CA}$ response to temperature as the red and orange lines. Without the $\alpha$ correction applied (Posterior with $\alpha = 1400$; orange line), the optimized $g_{CA}$ resembles the fluctuations in the observations but there remains a bias in the amplitude. In contrast, when the optimized value of $\alpha$ is

included, the amplitude of $g_{CA}$ is improved (red line). Due to the different optimized $\alpha$ values in each phenological stage, the improvement of the red line shows the appropriate temperature responses. For instance, in Harvard Forest, $\alpha$ in leaf-out and growth (1798 and 1740) mostly corresponds to lower temperatures. At these stages, the impact on $g_{CA}$ is smaller because $g_{CA}$ is smaller than that at high temperature. At high temperatures there are more significant corrections of $g_{CA}$, which correspond to the maturity stage value of $\alpha$ (2224).


The temperature responses of $g_{CA}$ and COS leaf uptake now show an optimum temperature in Hyytiälä, characterized by the parameter $T_{eq}$ ($T = 295$ K (22 ℃)). In addition, the improvement is significant at temperatures both below and above the optimum temperature in Hyytiälä. The optimized $g_{CA}$ is larger throughout the whole temperature range in Harvard Forest,





without an optimum within the measured range. This corresponds to $T_{eq} = 311\ K\ (38\ °C)$, a value substantially higher than
found at Hyytiälä.



**Figure 7. Temperature dependency on $g_{CA}$ (a) and COS leaf uptake (b) in Hyytiälä (HYYT, left) and Harvard Forest (HVFM, right). The lines are medians and the filled area represents the 25 to 75 percentiles of each temperature range with 3 °C intervals. Black: data based on observations; blue: previous parameters with $f(T_{can})_{SiB4}$; red: optimized parameters with $f(T_{can})_{new}$ and $g_s$ parameters of the BB model; orange: same as the red line but now $\alpha$ is prescribed with original value (1400) and not optimized.**





### 3.5 Application in SiB4

**3.5.1 Stationary monthly diurnal variation**

Figures 8 and 9 display the SiB4 simulation results obtained with the original and optimized parameterizations compared to observations for Hyytiälä and Harvard Forest, respectively. As a result of the optimization, the monthly diurnal variation of the optimized COS vegetation flux, $g_s$, and $g_{CA}$ are closer to observations than the original SiB4 simulations. The observed COS leaf uptake and $g_s$ show diurnal and seasonal fluctuations at both measurement sites, with the highest values around

midday and in summer. For $g_{CA}$, we observe a weak diurnal cycle throughout the year, and higher daytime maximum values in summer, driven by the temperature dependence of CA.

In Hyytiälä, COS leaf uptake in the original SiB4 model was underestimated during daytime in all months. The fluxes increased too slowly in the morning for all months (Fig. 8(a)). These issues are solved by optimizing the BB model parameters and

temperature response function. In the case of $g_s$ (Fig. 8(b)), the original SiB4 simulation showed the correct timing of the increase and decrease of $g_s$ in the morning and afternoon, but underestimated the peak daytime values. The optimized model now better resembles the daytime $g_s$ values.

However, the model still overestimates $g_s$ in the late afternoon of summer months at Hyytiälä. We speculate that one of the

reasons lies in an inaccurate humidity, or humidity stress in SiB4. Figure 10 shows a diurnal cycle of $g_s$ simulations averaged from April to August with different choices on how the humidity stress factor is treated (Sect. 2.5). When the default 0.7 threshold of humidity stress ($F_{LH}$) in ENF is applied in SiB4, $g_s$ is overestimated in the afternoon in Hyytiälä (blue dotted line). When we removed the minimum threshold of $F_{LH}$ for ENF, $g_s$ simulations during mid-day are improved (note that the threshold was only implemented for ENF, not for DBF, and thus the blue dotted line is not visible for Harvard Forest in Figure 10).

However, SiB4 still tends to overestimate $g_s$ in the morning and late afternoon. In contrast, when we base the $g_s$ calculation on the observed RH above the canopy, the diurnal cycle is better simulated (orange dashed line). This implies that SiB4 has the tendency to underestimate the humidity stress in the late afternoon when converting observed specific humidity above the canopy to humidity at leaf surface level.

The optimized model still significantly underestimates $g_s$ at Hyytiälä in April, September, and October (Fig. 8b). This might indicate that we did not properly separate stomatal transpiration rates from the observed latent heat flux. The simulated mean ratios of evaporation to evapotranspiration in these three months are 66 %, 60 %, and 95 %, respectively, and these values are higher compared to the other months (43 to 53 %). Thus, we speculate that the observed evapotranspiration does not solely represent stomatal transpiration in these months, leading to overestimated $g_s$ in the observation.






Figure 8(c) shows that the optimized $g_{CA}$ often resembles the observed daytime $g_{CA}$ better than the original SiB4 simulation. Only in April is the optimized $g_{CA}$ overestimated in Hyytiälä. Again, this can likely be explained by the underestimated $g_s$, which is used to derive pseudo-observations of $g_{CA}$ (see Sect. 2.2.2).

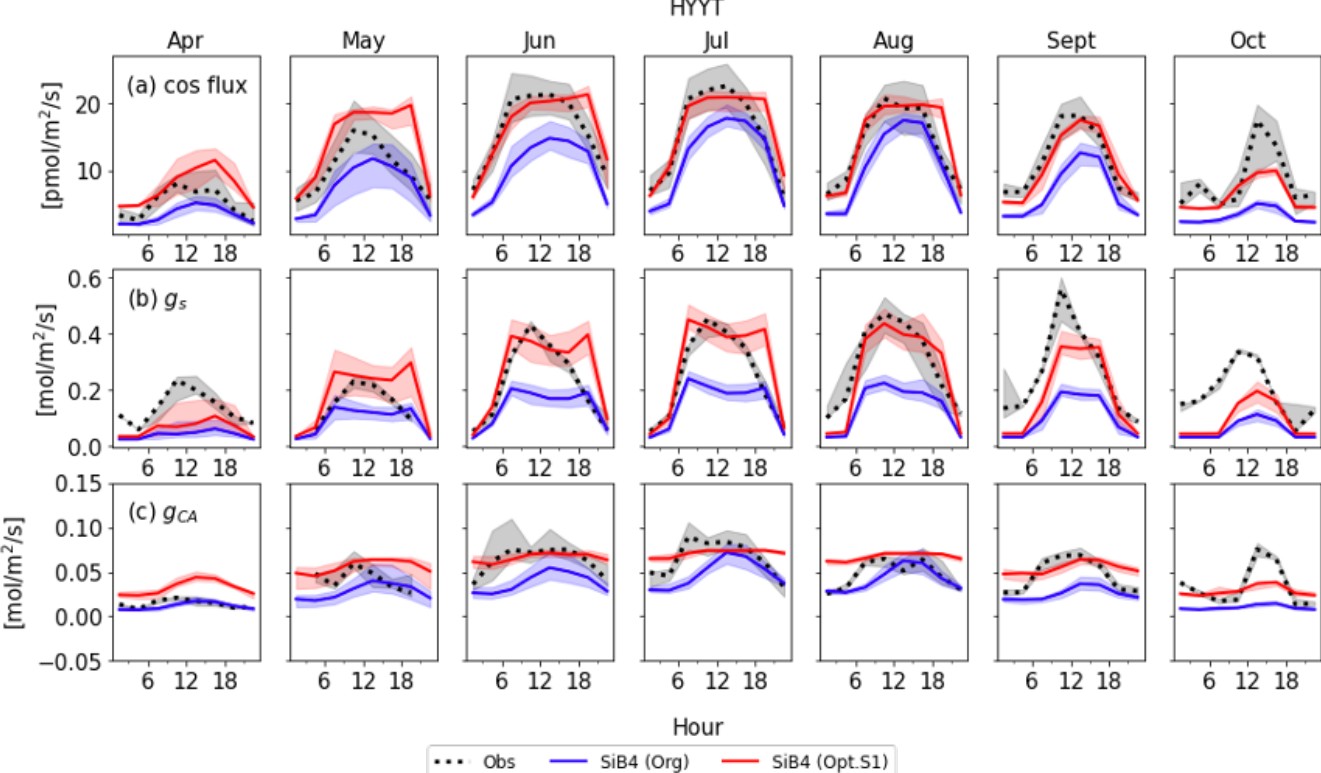

**Figure 8. Monthly-diurnal cycle of COS leaf uptake (a), $g_s$ (b), and $g_{CA}$ (c) in Hyytiälä (HYYT) from 2012 to 2016. Data include observations (black dots), the original SiB4 model with $f(T_{can})_{SiB4}$ (blue solid), and SiB4 with optimized $g_s$ and $g_{CA}$ parameters with $f(T_{can})_{new}$ (red solid). The filled area corresponds to the 25-75 percentile of the data in each three-hourly interval of each month.**


At the Harvard Forest, the optimized SiB4 model generally simulates the magnitude of the COS leaf uptake well (Fig. 9(a)). The model overestimates the COS leaf flux only in the afternoon during the summer months. However, $g_s$ values are generally overestimated and SiB4 simulates two peaks during daytime, indicating humidity stress only shortly at mid-day. However, in reality, the humidity stress likely remains a limiting factor in the afternoon under conditions with high vapor pressure deficit

(VPD). Observations show that $g_s$ typically peaks in the early morning and decreases in the afternoon due to higher afternoon VPD. Figure 10 shows that, similar to the Hyytiälä simulation, the afternoon decrease in $g_s$ at Harvard Forest is better simulated when we use the RH observed above the canopy. In Fig. 9(c), the optimized $g_{CA}$ during the daytime agrees well with the pseudo-observations, except for several drops or peaks in July and October, likely caused by observational errors or uncertainty of the observed $g_s$.



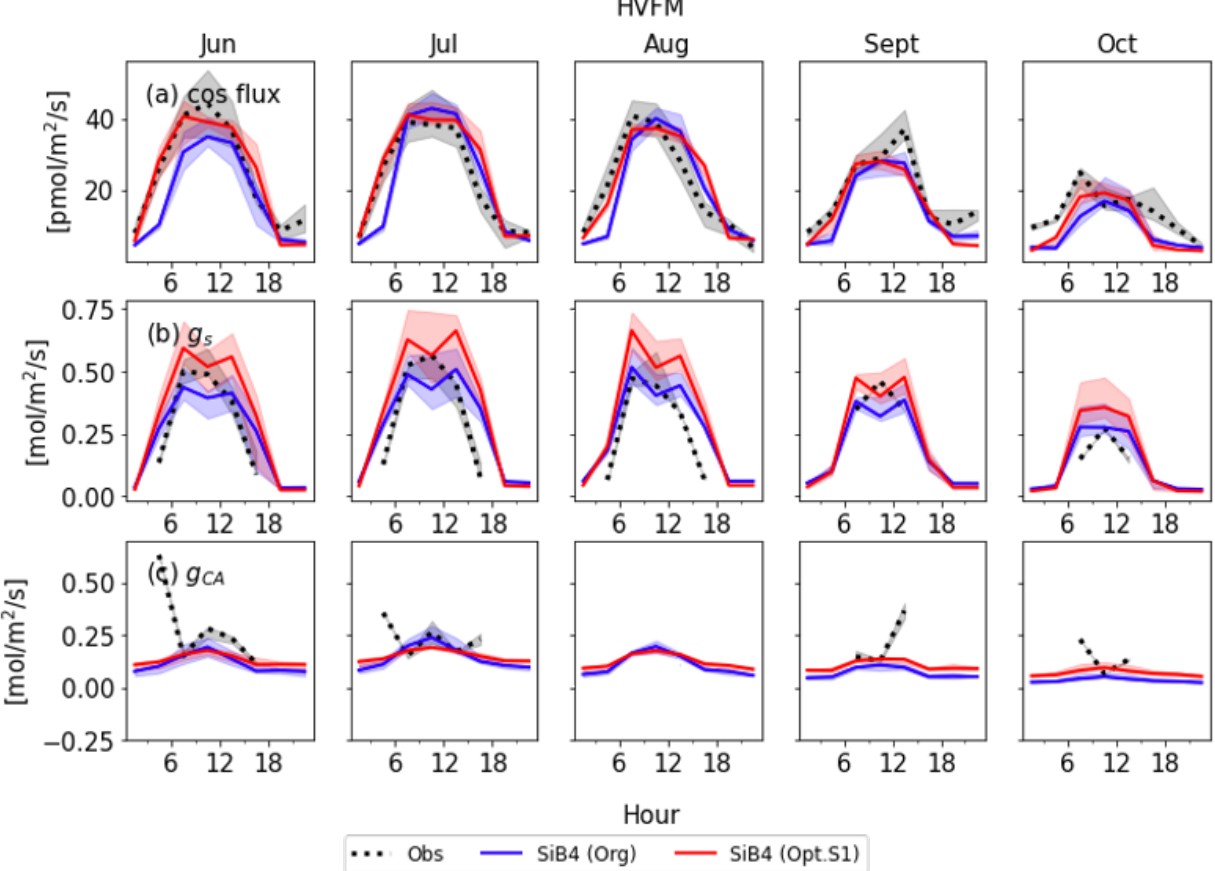

**Figure 9. Same as Fig. 8 but for Harvard Forest (HVFM).**




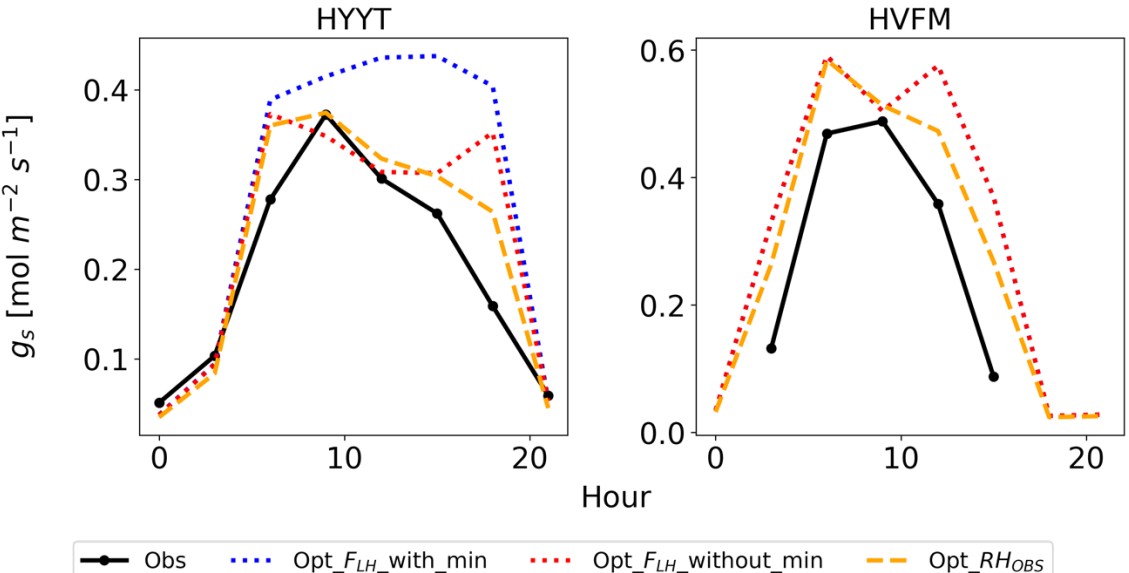

**Figure 10. Average diurnal cycle of $g_s$ in Hyytiälä (HYYT) and Harvard Forest (HVFM) from April to August. Data include observations (black solid), SiB4 simulation with optimized parameters with minimum bounds of $F_{LH}$ (blue dotted), without bounds (red dotted), and with observed RH in air (orange dashed). Note that the blue and red lines overlap for HVFM.**

### 3.5.2 Global application

Figure 11 shows the biosphere COS flux (soil and the optimized vegetation flux) using the optimized parameters and the difference with the original SiB4 model. In general, the COS biosphere uptake is lower in the tropics and higher towards high latitudes in the SiB4 model with optimized parameters. We find the same patterns for all seasons. The differences are consistent with canopy temperature variations. When temperatures are below 3 °C (boreal) and 3-25 °C (temperate), the optimized COS biosphere uptake is larger compared to the original simulation, corresponding with higher $g_{CA}$ values calculated by the new temperature function in Fig. 7. In contrast, temperatures above about 25 °C result in lower COS biosphere uptake in the optimized run, reflecting the reduced enzyme activity at high temperatures in the new temperature response function. Note here that we also found that the temperature response of CA is different in different climate zones. Since we do not have observations in the tropics, the calculated lower uptake in the tropics remains very uncertain. The higher uptake at high latitudes and lower uptake at the tropics are nevertheless consistent with inverse modelling results presented in Ma et al. (2021) and would help towards closing the COS budget. Still, however, the temperature response function and BB parameters are now based on measurements of only two sites in only two biomes. With more measurements over different vegetation types, these parameters could also be optimized for a wider range of ecosystems.





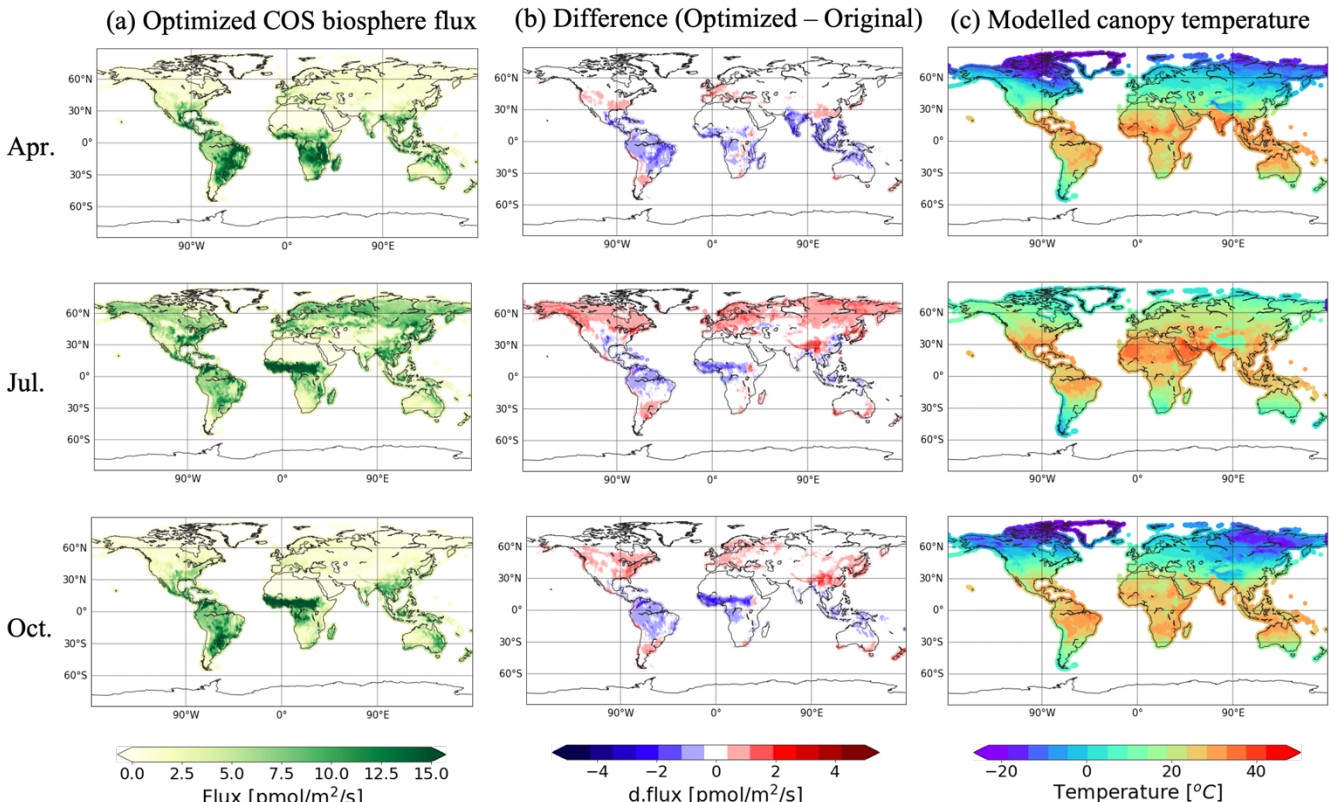

**Figure 11. Spatial distribution of optimized COS biosphere fluxes (a), the difference between optimized and original COS biosphere flux (b), and canopy temperature (c). All were estimated by the SiB4 model.**

## 4. Conclusion

525  To simulate more accurate COS leaf uptake in the SiB4 model, we have proposed a new temperature function $f(T_{can})_{new}$ for the CA enzyme and have optimized $g_s$ and $g_{CA}$ parameters using observations in ENF (Hyytiälä) and DBF (Harvard Forest) systems. The optimized model reduced the MBE from -4.65 to -0.34 $pmol\ m^{-2}\ s^{-1}$ and reduced $\chi^2$ from 1.01 to 0.61.

The new temperature function is characterized by an optimum temperature of 295 K (22 °C) (Hyytiälä) and 311 K (38 °C)

530  (Harvard Forest) with $\Delta H_a$ = 40 kJ mol⁻¹. The new function now considers an optimum temperature for enzyme activity, contrary to the initial temperature function used in SiB4 where an exponential increase of the temperature function was adopted from the RuBisCo enzyme activity. The new temperature response increases $g_{CA}$, and thereby the COS flux when the temperature is below the optimum temperature (mostly at high latitudes) and decreases the COS uptake at higher temperatures. (e.g. close to the equator). Globally, these modifications help to close gaps in COS budget that were identified in earlier studies.




In this study, we have interpreted the decreasing $g_{CA}$ at higher temperatures as an optimum enzyme activity with the widely applied assumption that there are no COS emissions in leaves. However, COS emissions have recently been reported at high temperatures (Maseyk et al., 2014, Commane et al., 2016, Gimeno et al., 2017). To determine reasons for reducing COS leaf flux and internal conductance at high temperatures, it will be necessary to analyze the possibility that leaf emissions exist in observations in the future.

We have optimized the BB model parameters for which we took advantage of the characteristics that the nighttime COS flux informs about nighttime $g_s$, and thus the parameter $b_0$. The improved correspondence between model and observations shows that COS observations can help to constrain the relation between $g_s$ and GPP better. In addition, we showed that SiB4 underestimates the leaf humidity stress under conditions where high VPD should limit $g_s$ in the afternoon. This can be improved with more accurate relative humidity values and removing the threshold of humidity stress that was implemented in SiB4 specifically for ENF.

The optimized parameters show different values depending on the PFT. Therefore, extending our approach with more observations in different climate zones and over different PFTs will help obtain accurate COS fluxes on a global scale. This approach would reduce the uncertainty in the global COS budget and provide additional constraints on GPP.

**Appendix A. State variable error settings**

To evaluate the impact of the various parameters in $f(T_{can})_{new}$ in the optimization as state variables, we implemented a sensitivity test of a total cost function combined with $cost_1$ and $cost_2$, excluding the background term in the cost function equation (Eq. (10)). Fig. A1 shows the shape of the cost function when one parameter is varied within an acceptable range, while the other parameters are fixed (Daniel et al., 2010; Sun et al., 2015) (details in Sect. 2.3.2). Based on the shape of the cost function, we used a pragmatic approach to select realistic parameter ranges. Variable values that push the cost function beyond 3.45 (Hyytiälä) and 5.14 (Harvard Forest) were considered outside the allowed physical range (red lines in Fig. A2). These thresholds are determined by the cost function value assuming that the modelled $H(x)$ is the 75-percentile value of observation in three-hourly observation in each month. Variables $\alpha$, $b_0$, $b_1$, and $T_{eq}$ (Fig. A1(a), (b), (c), and (f)) have more significant impacts on the cost function than $\Delta H_a$ (Fig. A1(d)) and $\Delta H_{eq}$ (Fig. A1(e)). Overall, costs in Harvard Forest are higher than in Hyytiälä, likely because DBF has larger diurnal and seasonal variations in the observed fluxes than ENF. We set the optimization range as an initial value ± 1.5 state error to apply SHGO algorithm.

Figure A2 shows contour diagrams of the cost function as a function of $T_{eq}$ and other parameters of $f(T_{can})_{new}$. The gradient is the cost function indicates the relative importance of each parameter. $\Delta H_{eq}$ does not interact with $T_{eq}$, but $\Delta H_a$ is inverse





proportional to $T_{eq}$ to minimize the cost. The cost function is most sensitive to variations in $T_{eq}$ and therefore we decided to fix $\Delta H_{eq}$ and $\Delta H_a$ at 100 kJ mol$^{-1}$ and 40 kJ mol$^{-1}$, respectively and to base our optimization on the state variables $\alpha$, $b_0$, $b_1$ and $T_{eq}$.

**Figure A1: Cost function values plotted against the value of the state vectors elements in Hyytiälä (solid line) and Harvard Forest (dotted line). The red lines indicate a criteria cost calculated by $H(x)$ as the 75-percentile value of every three-hourly observation in each month. While the target parameter changes, the other variables are fixed as $\alpha$ = 1400 (Hyytiälä), 2000 (Harvard Forest), $\Delta H_a$ = 40 kJ mol$^{-1}$ $\Delta H_{eq}$ = 100 kJ mol$^{-1}$, and $T_{eq}$ = 295 K (Hyytiälä), 310 K (Harvard Forest), $b_0$ = 0.02 (Hyytiälä), 0.01 (Harvard Forest), and $b_1$ = 17 (Hyytiälä), 12 (Harvard Forest). These values were decided where the cost has minimum.**





(a) $T_{eq}$ v.s. $\Delta H_{eq}$

(b) $T_{eq}$ v.s. $\Delta H_a$

**Figure A2: Contour diagram of the cost function value as a function of $T_{eq}$ and (a) $\Delta H_{eq}$ and (b) $\Delta H_a$ in Hyytiälä (left) and Harvard Forest (right). While the target parameter changes, the other variables are fixed as Figure A1.**

## Appendix B. Posterior uncertainties

To evaluate the ability of constrain the parameters, we performed an ensemble optimization with 40 different members. In each optimization, noise was added to the parameters ($\varepsilon_a$) and to the observation ($\varepsilon_y$). Random perturbations were drawn from a normal distribution with zero mean and standard deviations $\sigma_a$ for the state parameters and $\sigma_y$ for the observations. The new cost function of an individual optimization thus becomes:

$$J(x) = \frac{(x - x_a + \varepsilon_a)^2}{2\sigma_a^2} + \frac{\left(y + \varepsilon_y - H(x)\right)^2}{2\sigma_y^2} \tag{12}$$




We optimized each ensemble with the same observations (GPP and COS leaf uptake) and state variables but added noise to

each ensemble member (Chevallier et al., 2007). Subsequently, we calculated the posterior uncertainty as the one-standard

deviation of the posterior distribution of the optimized parameters.

Figure B1 shows the prior and posterior distribution of the parameters at the two stations. All posterior parameters show

considerable reductions of variations (error), with optimized values that are listed in the main text, Table 3.

Additionally, we calculated a correlation matrix between the posterior state parameters at the two stations, which is shown in

Fig. B2. Overall, each parameter does not interact significantly (covariances < 0.7) except for parameters $b_0$ and $b_1$. $b_0$ and $b_1$

influence the $g_s$ calculation in opposite ways. For example, a larger optimized value of $b_0$ corresponds to a smaller slope $b_1$.

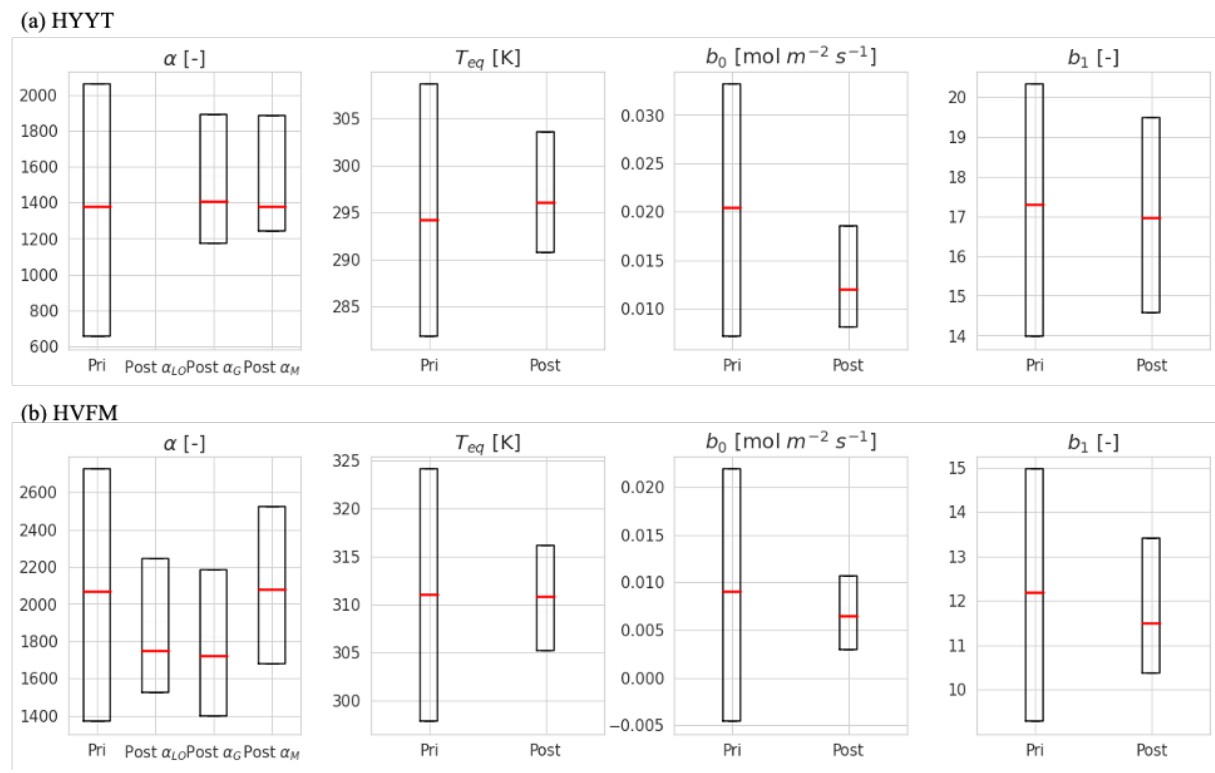


**Figure B1. Error reduction of state variables in two stations (Hyytiälä (HYYT) and Harvard Forest (HVFM)). The red lines represent median values, and the boxes represent errors. Column 'Pri' shows the initial value and state error. Column 'Post' represents the mean of the optimized state variables and the corresponding standard deviation. $\alpha_{LO}$, $\alpha_{G}$, $\alpha_{M}$ indicate $\alpha$ in each phenological stage (leaf-out, growth, and maturity, respectively).**






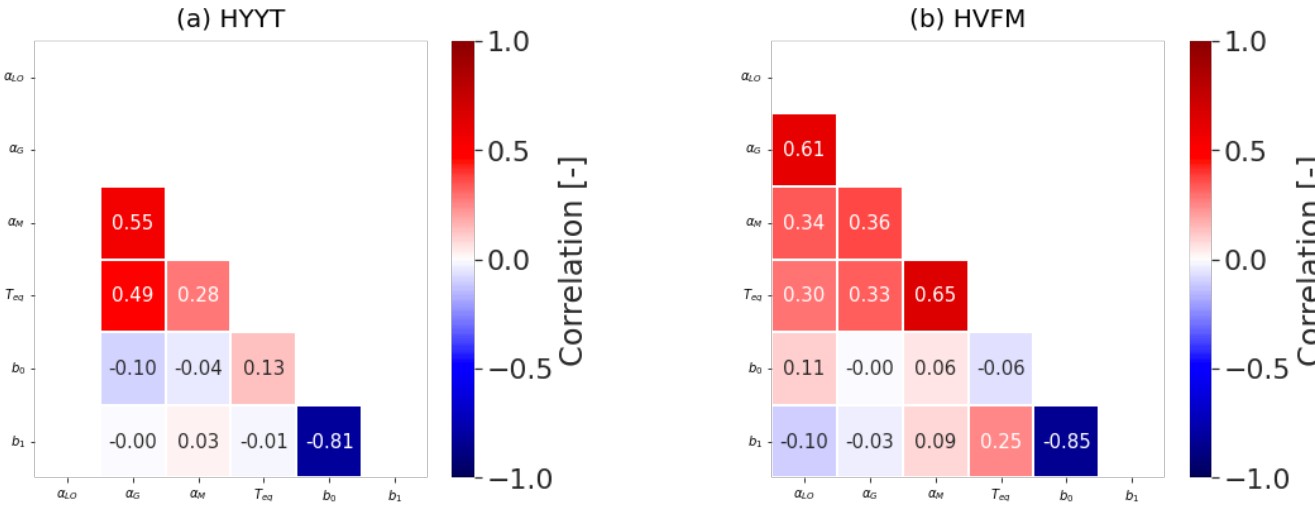

**Figure B2. Covariance matrix for all state variables in Hyytiälä (HYYT) and Harvard Forest (HVFM).**


**Code availability**

The SiB4 code is available online at https:// gitlab.com/kdhaynes/sib4v2_corral. TM5-4DVar model codes are available on the TM5-4DVAR website (https: //sourceforge.net/projects/tm5/).

**Author contributions**

AC, LMJK, and MK devised the study. AC optimized ecosystem parameters and analysed the results with consultation of LMJK and MK. KM and RW provided observation data and site-specific insights. AC wrote the manuscript and all authors provided comments.

**Competing interests**

The authors declare that they have no conflict of interest.


**Acknowledgements**

We thank everyone that contributed to the collection of data for Hyytiälä and Harvard Forest sites. The ecosystem dataset by eddy covariance at Hyytiälä was supported by ICOS Finland (319871) and the Atmosphere and Climate Competence Center



(ACCC) Flagship (Vesala et al., 2022). The soil dataset at Hyytiälä was collected from Sun et al. (2018). Data from Harvard
Forest is supported by the AmeriFlux Management Project (Wehr et al., 2017; Commane et al., 2015).

**Financial Support**

This work was funded through the ERC-advanced funding scheme (AdG 2016 Project Number: 742798, Project Acronym:
COS-OCS). This work was carried out on the Dutch national e-infrastructure with the support of SURF Cooperative. We
acknowledge computing resources from the Netherlands Organization for Scientific Research (NWO; grant no. NWO-
625 2021.010).

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
