# Peer review of "Optimizing the Carbonic Anhydrase temperature response and stomatal conductance of carbonyl sulfide leaf uptake in the simple biosphere model (SiB4)"

_EGUsphere, 2022_

## Author Comment (AC1)

Review #1

→ We appreciate the kind and valuable remarks and suggestions of the reviewer. We are very pleased to reflect on them and hope to convince the reviewer that the modified version of manuscript reads better.

We replied to your questions and remarks in blue, and modifications in the manuscript are expressed as underlined:

Initial work on the leaf COS uptake was based on the notion that the carbonic anhydrase (CA) conductance (gca) would be relatively large (or the corresponding resistance low) since CA is highly efficient in catalyzing COS. As a consequence, it was assumed that the leaf COS uptake would be mainly limited by stomatal conductance (gs), opening interesting avenues for using the leaf COS uptake as a proxy for transpiration and photosynthesis. By now more and more experimental data are surfacing which suggest that gca may be of similar magnitude as gs or even be the rate-limiting step for leaf COS uptake. There is thus an urgent need to better understand gca, both in terms of inter-specific differences and what these relate to, as well as with regard to the short-term drivers, and this information needs to be included in models which simulate the leaf COS uptake.

The manuscript by Cho et al. makes an important and timely contribution to this field by suggesting a peaked as opposed to the previous purely exponential temperature response of gca in the model SiB4. The updated model is able to reproduce the temperature response of the canopy-scale COS at two different forest study sites and in a global application the COS uptake is increased in higher latitudes and decreased in the tropics. In addition, the authors constrain the parameters of the stomatal conductance model inside SiB4 by means of the COS flux measurements.

Overall, most of my comments are minor, but there are many of these, aimed at improving the clarity of the writing, as summarized below.

The one, possibly, major, comment relates to the fact that the authors optimized parameters affecting the supply side of photosynthesis, i.e. the b1 stomatal parameter, against experimentally derived GPP, but not the demand side, e.g. Vcmax. I presume that all parameters the authors did not optimize, were left at the default values for the corresponding PFTs. This could mean that by optimizing the b1 parameter, the authors might have mapped differences between the (unknown) true and pre-scribed Vcmax into the b1 parameter. Furthermore, since gca is scaled to Vcmax, this might have further consequences for the estimated alpha parameter and possibly even the temperature reponse parameters of gca. I would like the authors to state why they did not choose to optimize some parameter representing the demand side of photosynthesis and discuss what the implications of not doing so might be. Ideally, they would underpin their arguments with some evidence which convincingly shows that any bias in Vcmax does not affect the parameters they retrieve and their interpretation.

→ We agree that it would be logical to also optimize Vmax of Rubisco. However, Vmax is hard to solve within the SIB4 framework. As written in the manuscript, in the first step we optimize b1 with experimentally derived GPP. SIB4 uses here the Ball-Woodrow-Berry equation (Eq. (2), Eq. (11)) in the COS uptake calculations, which does not need Vmax of Rubisco, but rather relates $g_s$ to GPP. Mapping the optimized b1 parameter to Vmax is difficult. A more direct coupling of $CO_2$ and COS could make this possible, but the SiB4 $CO_2$ model has many input factors, which complicates the optimization procedure. We are currently working on coupling $CO_2$ and COS within a simple model that we use to analyze laboratory observations, and we hope to report on this soon. Indeed, here we attempt to optimize also the demand side. We modified the manuscript as follows:

- In the manuscript
  (Original) *(lines 293-294 in the old version)*
  The $V_{max}$ of RuBisCo was found to vary over the phenological stage and per PFT (Woodward et al., 1995; Wolf et al., 2006; Kattge et al., 2009; Walker et al., 2014), which also affects the calibration factor $\alpha$. Therefore, we optimized $\alpha$ for each PFT and each phenological stage. In contrast, $b_0$, $b_1$, and $T_{eq}$ were only separately determined for the different PFTs, assuming local characteristics for each PFT.

  (Modified) *(lines 306-311 in the new version):*
  The $V_{max, rub}$ was found to vary over the phenological stage and per PFT (Woodward et al., 1995; Wolf et al., 2006; Kattge et al., 2009; Walker et al., 2014), which also affects the calibration factor $\alpha$. Therefore, we optimized $\alpha$ for each PFT and each phenological stage. In contrast, $b_0$, $b_1$, and $T_{eq}$ were only separately determined for the different PFTs, assuming local characteristics for each PFT. We did not include $V_{max, rub}$ in the state variables because this would require SIB4 $CO_2$ simulations. These simulations need several parameters, like carbon cycle pools, which are difficult to estimate. Therefore, we focus this research on estimating $V_{max, CA}$ by optimizing $g_i$-related parameters.

Also, we explain the reason for using the BWB model.

- In the manuscript
  (Original) *(lines 264-269 in the old version):*
  We select to use GPP observations in the optimization over $g_s$ because GPP can be used to evaluate $b_1$ and $b_0$ using the BB model. Moreover, observed GPP leads to more accurate $b_1$ values, because of uncertainties in GPP are smaller compared to observation-based $g_s$. Here, we use only positive $GPP_{obs}$ values (uptake) because the BB model is only applicable in daytime conditions. The estimated GPP ($H(b_1)$ in Eq. (10)) is calculated by rewriting the BB model using observation-based $g_s$ (Sect. 2.2.2), modelled RH at the leaf surface ($F_{LH}$), and simulated $CO_{2S}$ from SiB4. Hereinafter the estimated GPP by the BB model is called $GPP_{BB}$ :

(Modified) *(lines 276-281 in the new version):*
We select GPP for the first step optimization rather than $g_s$, because derived GPP from NEE has been evaluated more frequently than observation-based $g_s$. We use only positive $GPP_{obs}$ values (uptake) because our target parameter $b_1$ in the first step cannot be optimized when GPP is zero. Here, we do not use $GPP_{SiB4}$ because SiB4 does not apply the BWB model for GPP calculation as described in Eqs. 5–7. For this reason, we cannot optimize BWB parameters with $GPP_{SiB4}$. Instead, we estimated GPP by rewriting the BWB model using an observation-based $g_s$ (Sect. 2.2.2), modelled RH at the leaf surface ($F_{LH}$), and simulated $CO_{2S}$ from SiB4. Hereinafter the estimated GPP by the BWB model is called $GPP_{BWB}$ :

Finally, I would like to suggest, following Sun et al. (2022, 10.1111/nph.18178), to replace the term gca with gi as conceptually all conductances/resistances other than ga, gb and gs are mapped into gca, notably the mesophyll conductance.

→ Thank you for this suggestion. We changed $g_{CA}$ to $g_i$ throughout our manuscript.

Detailed comments:

1. 14: "… respond differently to temperature."

   → We changed the sentence as your suggestion.

2. 15: the original paper on this stomatal conductance model was written by Ball, Woodrow and Berry – I think we should not forget about co-author Woodrow and name the model accordingly (BWB) – here and anywhere else in the manuscript

   → We agree that the name should be changed from Ball-Berry (BB) to Ball-Woodrow-Berry (BWB).

3. 18: but the model is driven by Tcan not Tair …

   → We changed the sentence as your suggestion.

4. 19-22: all these numbers may be too much detail for the abstract

   → We removed the details of $b_0$ because the changes are too small.

   - In the manuscript:
     (Original) *(lines 19-22 in the old version):*
     Optimized values for the Ball-Berry offset parameter $b_0$ (ENF: 0.013, DBF: 0.007 mol m$^{-2}$ s$^{-1}$) are higher (lower) than the original value (0.010 mol m$^{-2}$ s$^{-1}$) in the ENF (DBF), and optimized values for the Ball-Berry slope parameter $b_1$ (ENF: 16.36, DBF: 11.43) are higher than the original value

(9.0) at both sites.

(Modified) *(lines 19-20 in the new version):*
Optimized values for the BWB offset parameter are similar to the original value (0.010 ± 0.003 mol m$^{-2}$ s$^{-1}$), and optimized values for the BWB slope parameter (ENF: 16.4, DBF: 11.4) are higher than the original value (9.0) at both sites.

5. 26: these gaps are poorly identified and it is also not shown how these new estimates help close these gaps

→ We removed the sentence and added the amount of global COS biosphere flux changes.

- In the manuscript:
(Original) *(lines 24-26 in the old version):*
Furthermore, we simulate global COS biosphere fluxes, which show smaller COS uptake in the tropics and larger COS uptake at higher latitudes, corresponding with the updates made to the CA temperature response. This SiB4 update helps resolve gaps in the COS budget identified in earlier studies.

(Modified) *(lines 24-27 in the new version):*
Furthermore, global COS biosphere sinks with optimized parameters show smaller COS uptake in regions where the air temperature is over 25 °C, mostly in the tropics, and larger uptake in regions where the temperature is below 25 °C. This change corresponds with reported deficiencies in the global COS fluxes, such as missing sinks at high latitudes and required sources in the tropics.

6. 34: during nighttime ecosystem respiration can be measured … the problem is during the day when there is both GPP and RECO, but only NEE can be measured

→ We added the phrase 'during daytime' in the sentence.

- In the manuscript:
(Original) *(lines 31-33 in the old version):*
Observations of the net ecosystem exchange (NEE) of $CO_2$ include both Gross Primary Production (GPP) and ecosystem respiration, and those two individual components cannot be directly observed.

(Modified) *(lines 33-35 in the new version):*
Observations of the net ecosystem exchange (NEE) of $CO_2$ include both Gross Primary Production (GPP) and ecosystem respiration, and those two individual components cannot be directly observed during daytime.

7. 39: gs is seldomly derived from NEE for many reasons; typically the H2O flux would be used, which has problems as well (which you discuss later); if the internal conductance to COS is known (aka gca), then COS fluxes in principle would allow estimating gs both during day and night

    → Thank you for your suggestion. To clarify the meaning, we changed the sentence as below.

    • In the manuscript:
    (Original) *(lines 37-39 in the old version):*
    Therefore, measurements of COS uptake can provide information on stomatal conductance, e.g. during the night (Kooijmans et al., 2017), which cannot be obtained from CO2 measurements.

    (Modified) *(lines 39-40 in the new version):*
    Therefore, when the CA activity is accurately quantified, measurements of COS uptake can provide information on stomatal conductance (Kooijmans et al., 2017).

8. 69: here or in the next section it would be useful to review what is known about the temperature response of CA from physiological studies

    → Thank you for this suggestion. We added another study about the different temperature response between RuBisCo and CA in the paragraph.

    • In the manuscript:

    (Modified) *(lines 61-74 in the new version):*
    In SiB4, the COS assimilation is described as a series of resistances (i.e. inverse conductances) at the leaf boundary layer ($g_b$), the stomatal pores ($g_s$), and the leaves' interior ($g_i$). The $g_b$ and $g_s$ of COS are scaled relative to conductances for water vapor or $CO_2$ with diffusivity ratios and a calibration factor. For $g_i$, previous studies found that both the CA enzyme activity (Badger and Price 1994) and mesophyll conductance (Evans et al., 1994) scale with the maximum velocity of carboxylation by the enzyme RuBisCo ($V_{max, rub}$). Therefore, the COS internal conductance in SiB4 is scaled to $V_{max, rub}$ through a single calibration factor $\alpha$ based on laboratory leaf gas exchange measurements (Stimler et al., 2010, 2011, Berry et al., 2013). However, the enzymatic control of COS and $CO_2$ assimilation differ. COS molecules are hydrolyzed by the enzyme CA in the mesophyll cells (Protoschill-Kreb et al., 1996). In contrast, photosynthesis is further controlled by the enzyme RuBisCo. Thus, $CO_2$ has a different point of uptake compared to COS. The enzyme activity depends on the enzyme abundance and is related to environmental parameters such as temperature and pH (Michaelis and Menten, 1913). In particular, the CA enzyme does not require light to catalyze COS hydrolysis, whereas the RuBisCo enzyme does require light (Stimler et al., 2010). Different temperature responses of RuBisCo and CA were reported by Boyd et al. (2015) with the C4 plant *Setaria viridis.* They measured that

$V_{max,\ rub}$ increased with temperature in the range 10 to 40 °C, whereas the CA activity decreased above 30 °C. Currently, however, there is limited information about the temperature response function of CA.

9.  75-77: this could be actually be formulated as a hypothesis, giving the study a hypothesis-driven twist.

    → Since this study focuses primarily on COS, we refrain from using this suggestion in this manuscript.

10. 90: why are multi-year measurements need to constrain the model parameters?

    → This helps us to distinguish the valid signal from the noise. We slightly modified the manuscript as:

    - In the manuscript:
      (Original) *(lines 90-93 in the old version):*
      Several multi-year measurement datasets of $CO_2$ and COS biosphere and soil fluxes are now available (Commane et al., 2015; Wehr et al., 2017; Vesala et al., 2022), making it possible to use COS to provide information on $g_s$ and thereby constrain the BB model parameters.

      (Modified) *(lines 95-98 in the new version):*
      Several multi-year measurement datasets of $CO_2$ and COS biosphere and soil fluxes are now available (Commane et al., 2015; Wehr et al., 2017; Vesala et al., 2022). Multi-year datasets make it possible to distinguish valid signals from noise and to use COS to provide information on $g_s$ and constrain the BWB model parameters.

11. 95: with "observation-based gs" you apparently try to express that gs was not directly measured but inferred from measurements through some model; as this idea has not been introduced here yet, I suggest to formulate in a more unambiguous way; note that also GPP is not measured, but inferred through a flux partitioning model

    → Thank you for this suggestion. As you mentioned, the expression 'observation-based $g_s$' is ambiguous in this paragraph. We changed the sentence in the manuscript :

    - In the manuscript:
      (Original) *(line 95 in the old version):*
      To do so, we will use observed COS leaf fluxes and GPP, plus observation-based $g_s$.
      (Modified) *(lines 101 in the new version):*
      To do so, we will use eddy covariance (EC) measurements of the COS flux, GPP derived from NEE, and $g_s$ derived from the EC COS flux.

12. 103: remove "land" in "land surface energy"

→ We removed 'land'.

13. 105: it is unclear here how satellite information was used by SiB3 and how SiB4 differs – suggest to reformulate

→ We added the specific information of satellite data.

- In the manuscript:
  (Original) *(lines 104-106 in the old version):*
  Unlike the previous SiB3 model, which relies on satellite information, version 4 fully simulates the terrestrial carbon cycle using a process-based model (Haynes et al., 2019).

  (Modified) *(lines 111-113 in the new version):*
  Unlike the previous SiB3 model, which relies on satellite information to specify the time-varying phenological leaf state, version 4 fully simulates the terrestrial carbon cycle using a process-based model (Haynes et al., 2019).

14. 118: "… or conditions are unsuitable for photosynthesis."

→ We modified the sentence as you suggested.

15. 120: what about the aerodynamic resistance/conductance – shouldn't this be included in Eq. 1? Worth mentioning that gca conceptually incorporates any conductances downstream of the stomatal one, e.g. also mesophyll

→ The leaf uptake of COS is considered from the laminar boundary layer to the chloroplast. Therefore, the aerodynamic conductance is not considered in equation 1. However, this conductance is applied to calculate the total biosphere flux to connect the mole fraction in the canopy air space to the atmosphere. We added the information in the manuscript.

- In the manuscript:
  (Modified) *(lines 127-134 in the new version):*
  SiB4 simulates COS vegetation assimilation as a combination of three conductances from the laminar boundary layer to the chloroplast ($g_b$, $g_s$, and $g_i$) multiplied by the atmospheric COS mole fraction (Berry et al., 2013):

  $$F_{COS} = C_{COS} \left( \frac{1.94}{g_s} + \frac{1.56}{g_b} + \frac{1}{g_i} \right)^{-1} \qquad (1)$$

  where $F_{COS}$ is the COS vegetation assimilation in the canopy (pmol m$^{-2}$ s$^{-1}$), and $C_{COS}$ is the COS mole fraction in the canopy air space (pmol mol$^{-1}$). The factors 1.94 and 1.56 account for the smaller diffusivity of COS with respect to H$_2$O through the boundary layer and stomatal pores, respectively (Seibt et al., 2010; Stimler et al., 2010). Note that $g_i$ includes all conductances downstream of the stomata, such as the mesophyll conductance. Within SiB4, the aerodynamic conductance is used to connect the mole fraction in the canopy air space to the atmosphere.

16. 124-124: "The factors 1.94 and 1.56 account for the smaller diffusivity of COS with respect to H2O through the boundary layer and stomatal pores, respectively."

   → We modified the sentence as you suggested.

17. 135: "… the drought response …"

   → We modified the sentence as you suggested.

18. 141, "… most PFTs, but …"

   → We modified the sentence as you suggested.

19. 144: "… using the carbon pool …" – unclear what is meant here – isn't photosynthesis simulated as the minimum of Rubisco, light or storage-export limitation carboxylation rate?

   → Yes, the SiB4 model calculates photosynthesis as the minimum of the limitations by Rubisco, light, and storage export. We removed 'using the carbon pool with', which is ambiguous, and added the 'minimum' in the sentence. In addition, we removed the sentence 'The SiB4 model defines GPP as the minimum of three limiting rates' in the following paragraph.

   - In the manuscript:
     (Modified) *(lines 153-162 in the new version):*
     $GPP_{SiB4}$ is explicitly calculated in SiB4 as the minimum of three assimilation rates limited by enzyme activity ($w_c$), light ($w_e$), and carbon compound export ($w_s$) (Haynes et al., 2020). The three rates are calculated by functions $f_{c,e,s}$ described in detail in Sellers et al. (1996a) depending on a canopy temperature ($T_{can}$, K):
     $$w_c = f_c(V_{max}(T_{can}), pCO_{2i}, pO_2(T_{can}), \gamma^*) \qquad (3)$$
     $$w_e = f_e(APAR, pCO2_i, \gamma^*) \qquad (4)$$
     $$w_s = f_s(V_{max}(T_{can}), F_{RZ}, pO_2(T_{can})) \qquad (5)$$
     Where $pCO_{2i}$ (Pa) is the internal partial pressure of $CO_2$, $pO_2(T)$ (Pa) is the temperature response of partial pressure of $O_2$, APAR (mol m$^{-2}$ s$^{-1}$) is the absorbed photosynthetically active radiation, and $\gamma^*$ (Pa) is the $CO_2$ photo-compensation point. Note that $GPP_{SiB4}$ is used in SiB4 to calculate the COS leaf flux via $g_s$, as described in Eq. (2) and evaluated independently from GPP calculated by the BWB model ($GPP_{BWB}$), which will be introduced in Sect. 2.3.1.

20. 151-152: repetition from above

   → We removed the sentence "The SiB4 model defines GPP as the minimum of these three limiting rates." as mentioned in No.19.

21. 162: using a leaf energy balance approach?!

→ Yes, it is. We added the information in the sentence.

- In the manuscript:
  (Modified) *(lines 172-173 in the new version)*:
  In SiB4, the canopy temperature $T_{can}$ is calculated from the temperature above the canopy using the leaf surface energy balance (Sellers et al., 1996b), and $T_{can}$ is normally obtained from a meteorological analysis dataset.

22. 163: "air temperature"

→ We added the word in the sentence as you suggested.

23. 178: correct – actually very often also an optimum temperature response function is used for Vcmax and Jmax

→ For more informative texts, we added examples of using the Arrhenius-type equation in Vmax and Jmax

- In the manuscript:
  (Modified) *(lines 191-193 in the new version)*:
  The Arrhenius equation has been used for $V_{max, rub}$ and maximum rate of photosynthetic electron transport to estimate GPP (e.g. Dreyer et al., 2001; Galmés et al., 2016).

24. 200: "Observations"

→ We added the word in the sentence as you suggested.

25. 203, 206: GPP is not "observed", but derived from flux partitioning, i.e. a model

→ We changed the expressions in the related paragraphs:

- In the manuscript:
  (Original) *(lines 202-206 in the old version)*:
  In the optimization of $g_s$ and $g_{CA,}$ we used observed values of the variables required in the COS leaf uptake calculation (Eq. (1)), namely: COS ecosystem flux, COS soil flux, GPP, $C_{cos}$, temperature, and specific humidity. The observations were obtained at Hyytiälä in Finland during 2013-2017 (Kooijmans et al., 2017; Sun et al., 2018; Vesala et al., 2022) and at the Harvard Forest in the United States during 2012 and 2013 (Commane et al., 2015; Commane et al., 2016; Wehr et al., 2017). COS and GPP ecosystem fluxes were measured with the eddy-covariance (EC) technique.

  (Modified) *(lines 212-217 in the new version)*:
  In optimizing the parameters $g_s$ and $g_{i,}$ we used the following variables obtained from observation to calculate COS leaf uptake (Eq. (1)): the COS ecosystem flux, the COS soil flux, $C_{COS}$, temperature, specific humidity,

and GPP partitioned from NEE measurements. These data were collected and derived at Hyytiälä in Finland during 2013-2017 (Kooijmans et al., 2017; Sun et al., 2018; Vesala et al., 2022) and at Harvard Forest in the United States during 2012 and 2013 (Commane et al., 2015; Commane et al., 2016; Wehr et al., 2017). To validate the optimization results, we used the observation-based $g_s$ and $g_i$ (Sect. 2.2.2).

26. 209-210: what you mean is probably that the COS flux was calculated as the sum of the vertical eddy covariance and the storage flux – this is not a correction but required whenever the storage flux contributes significantly to the 3D mass balance

    → Thank you for your remark. We changed the word from 'corrected' to 'included' in the sentence.

    - In the manuscript:
      (Modified) *(lines 220-221 in the new version):*
      The effect of storage in the canopy airspace was included by collocated COS profiles (Kooijmans et al., 2017; Kohonen et al., 2020).

27. 211: why didn't you use GPP derived from CO2 flux partitioning as at Hyytiälä? This peculiarity might be should be further discussed given that it yields very different estimates compared to CO2 flux partitioning at HF

    → We chose the GPP from the isotope spectrometer for Harvard Forest, which has been reported as a more accurate method compared to NEE partitioning (Wehr et al., 2016). The isotopic flux partitioning (IFP) method partitions individual flux measurements. This method therefore reduces errors and biases by accounting for changes in the flux tower sampling footprint (Wehr and Saleska, 2015). Since in situ isotope measurements are limited, the IFP method is not applied to the Hyytiälä data. However, we want to take advantage of the IFP method for Harvard Forest, where the method was used for multiple years.

    Differences between the two methods are relatively small, as shown in Figure 1. However, when we applied the traditional GPP from NEE partitioning at Harvard Forest, RMSE, MBE, and $\chi^2$ deteriorated as shown in Table 1.

    We therefore prefer to use the IFP GPP at Harvard Forest.

[Figure]

Figure 1. Monthly and hourly variation of median values of GPP in each three-hourly period at Harvard Forest. The black line is GPP derived from the NEE partitioning method and red line is GPP obtained from the isotopic partitioning method.

| Station | Type | $g_i$ f(T) | RMSE $(pmol\ m^{-2}\ s^{-1})$ | MBE $(pmol\ m^{-2}\ s^{-1})$ | $\chi^2$ |
|---|---|---|---|---|---|
| Harvard Forest | Previous SiB4 | | 10.45 | -2.53 | 1.19 |
| | Posterior with isotopic partitioning GPP | $f(T)_{SiB4}$ | 9.54 | -1.99 | 0.81 |
| | Posterior with NEE partitioning GPP | | 9.67 | -2.27 | 0.83 |

Table 1. RMSE, MBE, and $\chi^2$ for the estimation of *COS leaf uptake* at Harvard Forest using prior stomata parameters (Pri) and posterior parameters using isotope-partitioning GPP and NEE-partitioning GPP.

Reference:
Wehr, R, and S.R. Saleska. "An Improved Isotopic Method for Partitioning Net Ecosystem–Atmosphere CO2 Exchange." *Agricultural and Forest Meteorology* 214–215 (December 15, 2015): 515-531. https://doi.org/10.1016/j.agrformet.2015.09.009.

- In the manuscript:
  (Original) *(lines 208-212 in the old version):*
  For Hyytiälä, the EC processing steps were described by Kohonen et al. (2020) and Vesala et al. (2022) and GPP was derived from NEE using multi-year parameter fits (Kolari et al., 2014, Kohonen et al., 2022). The effect of storage in the canopy airspace was corrected by collocated COS profiles (Kooijmans et al., 2017; Kohonen et al., 2020).
  For the Harvard Forest, we used GPP derived from $CO_2$ isotope EC measurements as reported in Wehr et al. (2016), and we used canopy COS uptake derived from COS EC measurements as reported in Wehr et al. (2017).

  (Modified) *(lines 219-224 in the new version):*
  We used canopy COS uptake derived from COS EC measurements for Hyytiälä (Kohonen et al., 2020; Vesala et al., 2022) and Harvard Forest (Wehr et al., 2017). The effect of storage in the canopy airspace was included by collocated COS profiles (Kooijmans et al., 2017; Kohonen et al., 2020).
  GPP at Hyytiälä has been obtained from NEE using multi-year parameter fits (Kolari et al., 2014, Kohonen et al., 2022). For Harvard Forest, we chose to use the GPP derived from the isotope spectrometer measurements, because it is more accurate and reliable with frequent and rigorous calibrations (Wehr et al., 2016).

215: averaging does not improve "data quality", all it does it reduces variability due to random uncertainty, but not the systematic one

→ Thank you for pointing this out. Actually, the median data were picked as the representation for three-hourly data, not averaged data. We modified the text.
- In the manuscript:
  (Original) *(lines 215-216 in the new version):*
  To ensure data quality for the COS ecosystem flux, soil flux, GPP, and COS mole fraction, we used three-hourly averages each month for each observed variable.

  (Modified) *(lines 230-231 in the new version):*
  To convert the data frequency of observations to SiB4's three-hourly time resolution, we calculated the median value of each variable in each three-hourly interval and for each month.

28. 216: that means you excluded 50 % of the data in each 3-hour period?!

→ We choose the median value every three hours to compare with the SiB4 model output which has an interval of three hours. We removed the phrase about the interquartile range and corrected the expression in the paragraph:

- In the manuscript:
  (Original) *(lines 215-218 in the new version):*
  To ensure data quality for the COS ecosystem flux, soil flux, GPP, and COS mole fraction, we used three-hourly averages each month for each observed variable. We removed outliers that fell out of the 25 to 75 percentile range in each three-hourly period. We only used data points when more than three data points were present at three-hourly time intervals in each month and when all variables required for the optimization were available.

  (Modified) *(lines 230-232 in the new version):*
  To convert the data frequency of observations to SiB4's three-hourly time resolution, we calculated the median value of each variable in each three-hourly interval and for each month. We only used data points when more than three data points were present and when all variables required for the optimization were available.

221-223: this sentence applies only to Hyytiälä?!

→ Yes, this sentence explains the data at Hyytiälä only. To avoid confusion, we changed the paragraphs like:

- In the manuscript
  (Original) *(lines 224-235 in the old version):*
  Figure 2 shows the resulting average diurnal cycle per month for COS ecosystem, soil, and vegetation fluxes (ecosystem flux minus soil flux). Note that positive fluxes indicate uptake. As the seasonal and diurnal variations in COS soil fluxes were small (Sun et al., 2018), we applied the monthly average diurnal cycle of the soil flux from 2016 to the other years (2013-2015 and 2017).

  (Modified) *(lines 225-234 in the new version):*
  COS soil flux measurements were available for the 2016 growing season at Hyytiälä, and for the 2012 and 2013 growing seasons at Harvard Forest. For the soil flux in other years at Hyytiälä, we applied the monthly average diurnal cycle of the soil flux from 2016 to the other years (2013-2015 and 2017). The seasonal and diurnal variation of the soil flux is small compared to the total ecosystem uptake of COS (Sun et al., 2018). Hence, the averaged value of 2016 can be safely used for other years.

  To convert the data frequency of observations to SiB4's three-hourly time resolution, we calculated the median value of each variable in each three-hourly interval and for each month. We only used data points when more than three data points were present and when all variables required for the optimization were available. Figure 2 shows the resulting average diurnal cycle per month for COS ecosystem, soil, and vegetation fluxes (ecosystem flux minus soil flux). Note that positive fluxes indicate uptake. Again, we note that we use the averaged soil flux at Hyytiälä, because its variability is much smaller than the leaf flux.

227: are these 25-75% before or after filtering for the 25-75% range?

→ Do you mean the 25-75% of data in the interquartile range (25-75 percentile)? Figure 2 data indicate 25-75 percentile of whole dataset.

- In the manuscript:
  (Modified) *(lines 238-240 in the new version):*
  Figure 2. Monthly diurnal variation of COS fluxes in 2016 at Hyytiälä (a) and 2012-2013 at Harvard Forest (b, c). Lines are median values of each three hours period and filled areas indicate the interquartile range (25 to 75 percentile). Black: COS ecosystem flux, blue: soil flux, red: vegetation flux estimated as ecosystem minus soil flux.

29. 234-238: by now much more elaborate algorithms are available for T/ET partitioning – see Nelson et al. (2020, 10.1111/gcb.15314) – there are also packages for easy application

→ Thank you for your suggestions. In this paper, we adhere to using the FG method by Wehr and Saleska (2021) and Wehr et al. (2017). The other T/ET partitioning methods by Nelson et al. (2020) have uncertainty in containing systematic biases of NEE partitioning. Therefore, there will be a limitation to optimize stomatal conductance parameters.

30. 248: does gb from SiB4 include the aerodynamic conductance?! Gb and Ga could be calculated from standard flux tower observations as done in the papers by Wehr et al.

→ $g_b$ is the conductance from the canopy to the canopy air space, and $g_a$, is the aerodynamic conductance. The latter is not included in the Equation 1, as it is applied as the prognostic equation in SiB4 to update the COS mole fraction in the canopy air space.

251: does that mean that you just retained data in the interquartile range?

→ Yes, it does. We want to use the data in the interquartile range (25 to 75 percentile). We changed the sentence as:

- In the manuscript:
  (Original) *(lines 250-251 in the old version):*
  We removed additional outliers outside the 25–75 percentile range of the $g_{CA}$ dataset.

  (Modified) *(lines 261-262 in the new version):*
  To avoid excessive noise, we only retained $g_i$ values in the interquartile range (25–75 percentile) of three hours for each month.

31. 262: a sequential two-step process is not simultaneous …

→ Indeed, it is not simulated simultaneously. We changed the word 'simultaneously' to 'sequentially'.

- In the manuscript:
  (Modified) *(lines 272-273 in the new version):*
  To optimize the $g_s$ and $g_i$ parameters, we intend to use the information from GPP and COS leaf uptake measurements sequentially.

32. 266-267: the BWB model is applicable also in the darkness – in this case gs will represent b0; the point rather is that GPP should be zero without light

→ Thank you for your remark. In the first step of the optimization, GPP is used to optimize $b_1$ (not $b_0$). Therefore, nighttime signals are not useful in the first step. We changed the sentence as below.

- In the manuscript:
  (Original) *(lines 266-267 in the old version):*
  Here, we use only positive $GPP_{obs}$ values (uptake) because the BB model is only applicable in daytime conditions.

  (Modified) *(lines 277-278 in the new version):*
  We use only positive $GPP_{obs}$ values (uptake) because our target parameter $b_1$ in the first step cannot be optimized when GPP is zero.

33. 311: what uncertainty does this statement refer to? Random – systematic? How would systematic uncertainty be taken into account with your approach of calculating the CV over 3-hourly periods?

→ An observation error is calculated by a three-hourly averaged coefficient of variance (CV) multiplied by a mean flux in a phenological stage. CV shows the extent of variability in relation to the mean and is independent of the mean value. Multiplication with the mean flux in a phenological stage would thus include the systematic uncertainty.

34. 325-326: why not also take the other environmental drivers as measured at the flux towers?

→ We aimed to use the MERRA drivers as a standard because all tests, development, and tuning of SiB4 are done with MERRA driver files. For example, we know that photosynthetic active radiation of MERRA is higher than observations, leading to model-observation biases, see also Figure S7 of Kooijmans et al., 2021. We do, however, want to use temperature and relative humidity from observations so that the two variables itself could be used consistently throughout the manuscript, without having to switch between data from MERRA to observations.

35. 332-333: what exactly does this mean? You used alpha, bo and b1 determined for ENF and DBF for these PFTs but used the standard values for all other PFTs?

→ Yes, it is correct. We changed the sentence as:

- In the manuscript:
  (Original) *(lines 332-333 in the old version):*
  As we found that all target parameters differ per PFT in Sect. 2.3.2, we applied these parameters only to ENF and DBF.

  (Modified) *(lines 350-353 in the new version):*
  As we found that all target parameters differ between ENF and DBF (Appendix A), the application of the optimized parameters to other PFTs will likely be incorrect. Hence, we applied the optimized parameters only to ENF and DBF, and used the standard values of SiB4 for the other PFTs.

36. 340: something wrong with this sentence

→ We changed the sentence.

- In the manuscript:
  (Original) *(line 340 in the old version):*
  we simulated the global COS leaf uptake without the 0.7 threshold of FLH for ENF.

  (Modified) *(lines 359-360 in the new version):*
  To account for the optimized humidity impact on the global COS leaf uptake, we simulated the global COS leaf uptake without the 0.7 threshold of $F_{LH}$ for ENF.

37. 358-359: please elaborate how/why this finding supports your two-step calibration approach

→ We edited the sentence because the expression is ambiguous. We changed the sentence to briefly mention that we focus on $g_s$ and $g_i$ conductances rather than on $g_b$.

- In the manuscript:
  (Original) *(lines 358-359 in the old version):*
  This finding supports this study's proposed two-step optimization process to improve $g_s$ and $g_{CA}$ (see Sect. 2.3).

  (Modified) *(lines 378-379 in the new version):*
  Therefore, to improve the accuracy of COS leaf uptake simulation

effectively, parameters of $g_s$ and $g_i$ are evaluated and optimized, and $g_b$ is kept to its standard value.

38. Table 1 and 2: are these statistics combined for both sites? Given that GPP was estimated in quite a different fashion at both sites, I suggest to split the statistics

→ Thank you for your suggestion. We split the statistics into two stations.

39. Figure 6: same question as above – are both sites combined? If so, I suggest to split

→ Thank you for your suggestion. We split the statistics into two stations.

40. 396-397: "Thus the different optimum temperatures reflect the adaptation of the enzyme's temperature response to the prevailing temperatures"

→ We changed the sentence as you suggested.

41. 397-398: since temperature is a key driver of the model anyway this should not be an issue – maybe rather say that accurate climate information is important?

→ Since temperature response is essential in the model, it is crucial to know the optimal value of variable $T_{eq}$ exactly. According to Lee et al., 2007, $T_{eq}$ reflects the climate of plants' habitation. We derived $T_{eq}$ only for two stations, but it should be obtained for each geolocation.
We changed the sentence to stress the role of the temperature.:

- In the manuscript:
  (Original) *(lines 396-398 in the old version)*:
  Thus, the optimum temperature reflects the temperature dependence of the enzyme and its adaptation to temperature (Lee et al., 2007). This indicates that regional climate information is important for the correct estimation of $g_{CA}$.

  (Modified) *(lines 420-422 in the new version)*:
  Thus, the optimum temperature reflects the temperature dependence of the enzyme and its adaptation to temperature (Lee et al., 2007). This indicates that regional temperature information is important for correctly estimating $g_i$ globally.

42. 399: for which sites/climates did Ogee et al. derive these values?

→ In the soil model of Ogee et al. (2016) an optimum temperature of 25 °C is used, which reflects the observed temperature dependent CA activity by Burnell and Hatch (1988) (See Figure 1(d) in Ogee et al., 2016). They observed the temperature response of CA on maize grown in a glasshouse between 20 and 30 °C. This temperature dependent CA

activity applied in the temperature range 0 to 17 °C, i.e. outside the optimum temperature. Thus, the optimum 25 °C in Ogee et al. (2016) is an extrapolation of the Arrhenius plot from the observations. Comparing with the soil model by Ogee et al. (2016) is therefore less appropriate, and we changed the reference in the manuscript from a soil model to direct observations of CA activity (Burnell and Hatch, 1988 ; Boyd et al., 2015).

- In the manuscript:
  (Modified) *(lines 422-427 in the new version):*
  The optimum temperature can be compared with other observations. For instance, Burnell and Hatch (1988) observed increasing CA activity with maize grown in a temperate temperature range from 20 to 30 °C, relative to a temperature of 17 °C. Thus, we can assume the optimal temperature lies above 17 °C. Another study by Boyd et al. (2015) observed the C4 plant *Setaria viridis* with a temperature of 28 °C / 18 °C day/night, and a reduced CA activity is suggested at temperatures above 25 °C. This optimum temperature falls between our values derived for Hyytiälä and Harvard Forest.

43. 401: "… reduced from the default value of 1400 …"

→ We changed the sentence as you suggested.

44. 407-408: this is not necessarily true as gi depends on both alpha and Vcmax and differences in the COS flux also depend on gs – that is to say that the differences in COS flux between both sites may also be due to other factors

→ We agree with your statement. As you mentioned, there are other factors to determine the flux magnitude. We removed the sentence "Since the observed COS leaf uptake in Harvard Forest is larger than in Hyytiälä, larger values of $\alpha$ are derived for Harvard Forest." in lines 407-408 in the original manuscript.

45. 410: similar to what?

→ We added the objective of 'similar to'.

- In the manuscript:
  (Modified) *(lines 436-437 in the new version):*
  The optimized results of the BWB model parameters $b_0$ are similar to the original values used in SiB4, but $b_1$ values are mostly higher.

46. 411-412: to put these results into perspective – if you were to go into the field and quantify nighttime stomatal conductance using a porometer I would presume these differences would be buried in the variability of the measurements; that is to say these differences are really small

→ Thank you for your remark. We added the word 'slightly' for $b_0$ changes. According to your suggestion, we removed the details of $b_0$ in the abstract.

- In the manuscript:
  (Modified) *(lines 436-438 in the new version):*
  The optimized results of the BWB model parameters $b_0$ are similar to the original values used in SiB4, but $b_1$ values are mostly higher. The parameter values $b_0$ for Hyytiälä (0.013 mol m$^{-2}$ s$^{-1}$) and Harvard Forest (0.007 mol m$^{-2}$ s$^{-1}$) are slightly changed compared to the initial value (0.010 mol m$^{-2}$ s$^{-1}$).

47. 419-422: this is really important information in my view!

→ Thank you for your remark. We repeat the error reduction in the abstract.

- In the manuscript:
  (Modified) *(lines 17-18 in the new version):*
  We find that CA has optimum temperatures of 20 °C (ENF) and 36 °C (DBF), which is lower than that of RuBisCo (45 °C), suggesting that canopy temperature changes can critically affect CA's catalyzation activity. Optimized values for the BWB offset parameter $b_0$ (ENF: 0.019, DBF: 0.013 mol m$^{-2}$ s$^{-1}$) and the slope parameter $b_1$ (ENF: 16.4, DBF: 11.4) are higher than the original value ($b_0$: 0.010 mol m$^{-2}$ s$^{-1}$, $b_1$: 9.0) at both sites. The optimization reduces prior errors on all parameters by more than 50 % at both stations.

48. 430: now you call these pseudo-observations? I suggest to use a consistent terminology throughout the manuscript

→ As you suggested, we changed 'pseudo-observations' to 'observation-based' in the manuscript.

49. 435: to emphasize this point the authors may want to add the number of measurements, e.g. in temperature bins, to Fig. 7

→ We added the number of observations in Fig. 7.

- In the manuscript:

(a) $g_i$

[Figure]

(b) $F_{COS}$

[Figure]

Figure 7. Temperature dependency on $g_i$ (a) and COS leaf uptake (b) at Hyytiälä (HYYT, left) and Harvard Forest (HVFM, right). The lines are medians and the filled area represents the 25 to 75 percentiles of each temperature range with 3 °C intervals. Black: data based on observations; blue: previous parameters with $f(T_{can})_{SiB4}$; red: optimized parameters with $f(T_{can})_{new}$ and $g_s$ parameters of the BWB model; orange: same as the red line but now $\alpha$ is prescribed with original value (1400) and not optimized. The numbers refer to the number of observations in each temperature bin.

50. 455: note sure I understand the "stationary" in the subheading

→ We removed the word "stationary" in the subheading.

- In the manuscript:
  3.5.1 Monthly diurnal variation

51. 458: were the "original" SiB4 simulations also tuned to the site data? If not, isn't there a mix of structural model differences and tuning affecting this comparison?

→ Here, "original" SiB4 uses the same site observations and structural model. The only difference between "original" and "prior/posterior" is the function of T in $g_i$ and the parameter values of $g_i$ and $g_s$.

52. 469-478: this merits further discussion I think; when the model overestimates gs because of FLH, this means that FLH, which is the relative humidity of the air in the boundary layer close to the leaf surface, is too large; because FLH = eb/esat(Tcan), there are two options for this to occur – (1) eb, which is the vapor pressure of the air in the boundary layer close to the leaf surface, is too large, which could be the case because transpiration (T) is too large or the boundary layer conductance too small since eb = esat(Tcan) - T*P/gb for water vapor transport across the boundary layer; or (2) Tcan, the temperature of the saturated water vapor in the leaf intercellular space, is too low, which would make esat small and thus increase FLH; it might also be mentioned that using RH from the reference height instead of RH at the leaf surface is conceptually wrong as stomata would sense moisture at the leaf surface and not above the canopy; here in turn it might also be mentioned that the use of RH in the BWB model has been critiqued since a long time as experiments show that stomata do not sense RH

→ It is indeed important to consider the reason for the overestimated $F_{LH}$ in SiB4. When we put three factors on the table to consider, as you mentioned: (1) water vapor flux in the boundary layer to the leaf surface is too large, (2) boundary conductance is too small, and (3) leaf surface temperature is too small. Considering (3), when we compare the temperature in the canopy air space to the observed air temperature around the canopy, they agree well (See Figure 2 below). Regarding (1), we speculate it would be the main reason because we found that the observed RH above the canopy explains the diurnal fluctuation of $g_s$ (orange dashed line in Fig. 10) for both stations. Further research should evaluate the boundary conductance later with observation.

[Figure]

Figure 2. Scatter plot between modelled canopy air space temperature and observed air temperature.

Since we calculated $g_s$ based on the BWB method (Eq. 2), we assumed that RH affects $g_s$. Using the canopy RH rather than the leaf surface RH is indeed a limitation of this study. We added in the manuscript:

In the manuscript:
(Original) *(lines 475-478 in the old version):*
However, SiB4 still tends to overestimate gs in the morning and late afternoon. In contrast, when we base the gs calculation on the observed RH above the canopy, the diurnal cycle is better simulated (orange dashed line). This implies that SiB4 has the tendency to underestimate the humidity stress in the late afternoon when converting observed specific humidity above the canopy to humidity at leaf surface level.

(Modified) *(lines 503-511 in the new version):*
However, SiB4 still tends to overestimate $g_s$ in the morning and late afternoon. The overestimated $F_{LH}$ in SiB4 can result from three factors: (1) an overestimated water vapor flux in the boundary layer to the leaf surface, (2) an underestimated boundary conductance or, (3) an underestimated leaf surface temperature. Since we do not have observations of the leaf surface temperature, we confirmed that the estimated canopy temperature has a tight 1 to 1 relation with the observed air temperature. We speculate that the main reason for the overestimated $F_{LH}$ is the uncertain water vapor flux. When we base the $g_s$ calculation on the observed RH above the canopy, the diurnal cycle is better simulated (orange dashed line in Fig.10). The overestimated water vapor pressure implies that SiB4 tends to underestimate the humidity stress in the late afternoon when converting observed specific humidity above the canopy to humidity at leaf surface level. We suggest evaluating the boundary conductance (point (2) above) with observations.

53. 480: "significantly" in a statistical sense?

→ We removed 'significantly' from the text.

54. 495: back up statement with reference

→ We added the reference.

55. 514-516: can you provide some numbers here on how much the new simulations would help resolving the differences?

→ We added the way to calculate COS biosphere sink in Set. 2.4 and the quantified results are introduced in Sect. 3.5.2 with the new figure 11. Since general results of the new figure is similar to previous Figure 11, we removed the old plot which showed spatial distribution of difference between the optimized and original COS biosphere flux and temperature. With the new figure, we inspected three regions: northern boreal, northern temperate, and tropic regions. Overall, the global COS uptake remained similar. However, COS leaf uptake is smaller in tropics and larger in boreal and temperate regions.

- In the manuscript:
  (Original) *(lines 506-514 in the new version):*
  Figure 11 shows the biosphere COS flux (soil and the optimized vegetation flux) using the optimized parameters and the difference with the original SiB4 model. In general, the COS biosphere uptake is lower in the tropics and higher towards high latitudes in the SiB4 model with optimized parameters. We find the same patterns for all seasons. The differences are consistent with canopy temperature variations. When temperatures are below 3 ℃ (boreal) and 3-25 ℃ (temperate), the optimized COS biosphere uptake is larger compared to the original simulation corresponding with higher $g_i$ values calculated by the new temperature function in Fig. 7. In contrast, temperatures above about 25 ℃ result in lower COS biosphere uptake in the optimized run, reflecting the reduced enzyme activity at high temperatures in the new temperature response function. Note here that we also found that the temperature response of CA is different in different climate zones. Since we do not have observations in the tropics, the calculated lower uptake in the tropics remains very uncertain.

  (Modified) : Sect. 3.5.2 *(lines 540-552 and 555-558 in the new version):*

[Figure]

**Figure 11. Monthly COS sink and averaged temperature in the mixed layer ($T_m$) over global (a) and specific regions (north boreal: b, north temperate: c, tropics: d) with the original (blue line) and the optimized (red line) SiB4 model. The grey bars represent the differences in COS sink between the original and the optimized model (right axis). The yellow bars are averaged temperatures in the mixed layer.**

Figure 11 shows the SiB4 calculated changes in the monthly COS biosphere flux after applying the optimized temperature function and stomatal parameters. The global COS sink remains almost preserved (Original: 701 Gg S/year, Optimized: 704 Gg S/year), but the regional budgets change significantly. For example, the optimized model estimates larger COS uptake for all seasons in boreal and temperate regions and smaller uptake in the tropics. These changes are explained by the new temperature function of $g_i$ in Fig. 7. However, since the new temperature function is based on only two observation sites in the boreal and temperate regions, the calculated uptakes need more verifications with observations obtained in other areas and in different climate conditions, such as the tropics.

To estimate the global impact of our findings, we performed a global simulation to evaluate COS leaf uptake estimated by the updated $g_s$ and $g_{CA}$ values. The atmospheric COS mixing ratio $C_{cos}$ were taken from optimizations using the TM5 chemical transport model (Ma et al., 2021; Kooijmans et al., 2021).

(Modified) Sect. 2.4 *(lines 347-350 in the new version)*:
To estimate the global impact of our findings, we performed a global SiB4 simulation from 2016 to 2018 to evaluate the influence of the new parameters on the monthly COS biosphere fluxes which are averaged for three years. The atmospheric COS mixing ratio $C_{COS}$ were taken from optimizations using the TM5 chemical transport model (Ma et al., 2021; Kooijmans et al., 2021). We used three-hourly $C_{COS}$ averaged over 2016 to 2018 by Kooijmans et al. (2021).

56. 529-530: move this sentence after the second one in this section?

   → Thank you for your suggestion. We move the sentence after the second one.

57. 537: Gimeno however studied bryophytes, which is quite different from the vascular plants which the PFTs in SiB4 mainly represent

   → Yes, there are little observations that point to emissions from vascular plants. We would like to leave the possibility open, and emissions will be discussed in our following paper.

---

## Author Comment (AC2)

Review #2

→ We appreciate your valuable remarks and suggestions. Below, we will reflect on them and we are convinced that the modified version of manuscript is improved thanks to your comments. We replied to your comments (black) in blue, and modifications in the manuscript are expressed as underlined.

We changed words '$g_{CA}$' to '$g_i$' and 'BB model' to 'BWB model' in the manuscript.

**General Comments**

This paper addresses some important factors that influence the modeling of carbonyl sulfide, with the goal of improving our ability to use it to estimate GPP. Plant-specific optimization of conductance parameters is a really useful way to approach model improvement. The authors used some nice measurement datasets at a couple of different forested sites and were able to demonstrate reduced model-obs mismatch with their optimized setup. Globally this also addresses some of the gaps pointed out by previous studies (e.g. missing sink at high latitudes). While there is still room for improvement, this is a good first step in improving our ability to model OCS. My comments are mostly minor or eliciting clarification. As with RC1, I was also hoping the authors would circle back to the impact on Vcmax and ways to potentially optimize that independently, but there is always the next manuscript!

**Specific Comments**

- Para from line 41: Soil emissions may also play a role in some specific regions (e.g. hot areas or agricultural fields). See refs cited in Ogee et al, Biogeosci 2016.

   → Thank you for your suggestion. We added the 'soil emission' in the sentence.

- In the manuscript:
  (Original) *(lines 41-42 in the old version):*
  Atmospheric COS mole fractions vary around 500 parts per trillion (ppt) and are primarily influenced by biosphere uptake, ocean emissions, and anthropogenic emissions (Kettle et al., 2002).

  (Modified) *(lines 42-44 in the new version):*
  Atmospheric COS mole fractions vary around 500 parts per trillion (ppt) and are primarily influenced by biosphere uptake, ocean emissions, and anthropogenic emissions (Kettle et al., 2002). Depending on the environmental conditions, soils can act as a COS source or sink (Maseyk et al., 2014; Whelan et al., 2016).

- Lines 44-45: More recently, Hu et al (PNAS 2021) also showed existence of this missing sink at higher latitudes.

   → We added the reference in the sentence.

- In the manuscript:
  (Original) *(lines 43-45 in the old version):*
  Moreover, Berry et al. (2013) showed that a sink is missing, or a source is overestimated at higher latitudes. These findings ask for careful evaluation of all sources and sinks, including the biosphere.

  (Modified) *(lines 46-47 in the new version):*
  Moreover, Berry et al. (2013) and Hu et al. (2021) showed that a sink is missing, or a source is overestimated at higher latitudes. These findings ask for careful evaluation of all sources and sinks, including the biosphere.

- Line 105: add the word 'prognostic' to SiB4 description

  → We added the word 'prognostic' in the first sentence of the paragraph.

- In the manuscript:
  (Original) *(The line 102 in the old version):*
  The SiB4 model is a land surface model that calculates the COS flux as described in Berry et al. (2013)

  (Modified) *(The line 109 in the new version):*
  The SiB4 model is a prognostic land surface model that calculates the COS flux as described in Berry et al. (2013).

- Line 202: technically, GPP is not an 'observation' but a 'derived/modeled quantity' so should not be included in this list of obs.

  → Thank you for your remark. We changed the subheading and expressions concerning GPP. We also rearrange the paragraph from site-based to variables-based: COS flux, GPP, and COS soil flux measurements.

- In the manuscript:
  (Original)
  2.2.1 Observed variables, 2.2.2 Observation-based $g_s$ and $g_{CA}$
  (Modified)
  2.2.1 COS flux, GPP, and mixing ratio, 2.2.2. Conductances $g_s$ and $g_i$'

  (Original) *(lines 202-214 in the old version):*
  In the optimization of $g_s$ and $g_{CA}$, we used observed values of the variables required in the COS leaf uptake calculation (Eq. (1)), namely: COS ecosystem flux, COS soil flux, GPP, $C_{cos}$, temperature, and specific humidity. The observations were obtained at Hyytiälä in Finland during 2013-2017 (Kooijmans et al., 2017; Sun et al., 2018; Vesala et al., 2022) and at the Harvard Forest in the United States during 2012 and 2013 (Commane et al., 2015; Commane et al., 2016; Wehr et al., 2017). COS and GPP ecosystem fluxes were measured with the eddy-covariance (EC) technique.
  For Hyytiälä, the EC processing steps were described by Kohonen et al. (2020) and Vesala et al. (2022) and GPP was derived from NEE using multiyear parameter fits (Kolari et al., 2014, Kohonen et al., 2022). The effect of storage in the canopy airspace was corrected by collocated COS profiles (Kooijmans et al., 2017; Kohonen et al., 2020).
For the Harvard Forest, we used GPP derived from CO2 isotope EC measurements as reported in Wehr et al. (2016), and we used canopy COS uptake derived from COS EC measurements as reported in Wehr et al. (2017).
In addition to the COS ecosystem fluxes, COS soil flux measurements were available for the 2016 growing season at Hyytiälä, and for the 2012 and 2013 growing seasons the Harvard Forest.

(Modified) *(lines 212-228 in the new version):*
In optimizing the parameters $g_s$ and $g_i$, we used the following variables obtained from observation to calculate COS leaf uptake (Eq. (1)): the COS ecosystem flux, the COS soil flux, $C_{COS}$, temperature, specific humidity, and GPP partitioned from NEE measurements. These data were collected and derived at Hyytiälä in Finland during 2013-2017 (Kooijmans et al., 2017; Sun et al., 2018; Vesala et al., 2022) and at Harvard Forest in the United States during 2012 and 2013 (Commane et al., 2015; Commane et al., 2016; Wehr et al., 2017). To validate the optimization results, we used the observation-based $g_s$ and $g_i$ (Sect. 2.2.2).

2.2.1 COS flux, GPP, and mixing ratio
We used canopy COS uptake derived from COS EC measurements for Hyytiälä (Kohonen et al., 2020; Vesala et al., 2022) and Harvard Forest (Wehr et al., 2017). The effect of storage in the canopy airspace was included by collocated COS profiles (Kooijmans et al., 2017; Kohonen et al., 2020).
GPP at Hyytiälä has been obtained from NEE using multi-year parameter fits (Kolari et al., 2014, Kohonen et al., 2022). For Harvard Forest, we chose to use the GPP derived from the isotope spectrometer measurements, because it is more accurate and reliable with frequent and rigorous calibrations (Wehr et al., 2016).
COS soil flux measurements were available for the 2016 growing season at Hyytiälä, and for the 2012 and 2013 growing seasons at Harvard Forest. For the soil flux in other years at Hyytiälä, we applied the monthly average diurnal cycle of the soil flux from 2016 to the other years (2013-2015 and 2017). The seasonal and diurnal variation of the soil flux is small compared to the total ecosystem uptake of COS (Sun et al., 2018). Hence, the averaged value of 2016 can be safely used for other years.

- Line 282: change wording to 'observation and observation-derived quantities' since 'GPPobs' and 'gs' are not direct observables but rather derived quantities.

→ As you suggested, we added the word 'observation-derived quantities'.

- In the manuscript:
  (Original) *(lines 282-284 in the old version):*
  Figure 3 specifies which observations are used in which step ($g_s$, $GPP_{obs}$, $F_{COS, obs}$, $C_{cos}$ highlighted as grey) and which variables are simulated by SiB4 (e.g. $T_{can}$, $F_{LH}$, $GPP_{SiB4}$, $CO_{2S}$, $g_b$).
  (Modified) *(lines 295-296 in the new version):*
  Figure 3 specifies which observations and observation-based quantities are used in each step ($g_s$, $GPP_{obs}$, $F_{COS, obs}$, $C_{COS}$ highlighted as grey) and which variables are simulated by SiB4 (e.g. $T_{can}$, $F_{LH}$, $GPP_{SiB4}$, $CO_{2S}$, $g_b$).

- Fig 4 comment: is the reason for missing hours in the HVFM All plot that there is no data for certain phenological stages (apart from growth and maturity for which you have data at all hours)?

→ Yes! The GPP data from 18h to 3h at Harvard Forest are missing for all phenological stages.

- Line 332-333: does this imply you used the new f(Tcan) estimations for forests and applied them to grasslands as well? That seems like it could cause additional problems.

→ Yes, grasslands might have a different temperature of the enzyme CA's activity. However, without observations we cannot say much. The necessary measurement for other vegetation types are mentioned in section 3.5.2. "With more measurements over different vegetation types, these parameters could also be optimized for a wider range of ecosystems."

- Line 344 comment: did you investigate whether 100 was sufficiently large?

→ To show the convergence of the optimized values, we now performed 100 simulations, and plotted the median posterior values against the number of simulations (Figure 1). Overall, we observe that the optimized median converged to their final values somewhere after ~100 optimizations.

[Figure]

Figure 1. Median values of state variables at the two stations as a function of the number of randomly perturbed simulations (HYYT: Hyytiälä; HVFM: Harvard Forest)

- Table 3: where does the prior error range come from? Perhaps a reminder is in order referencing Appendix A where the prior error is estimated (as mentioned in sec 2.3.2)

→ We agree with your suggestion. We added the related expression to the caption in Table 3.

- In the manuscript:
  (Original) *(lines 424-425 in the old version):*
  Table 3. Original (Org) and optimized (Post) state vectors for Hyytiälä and Harvard Forest in different phenological stages as defined by SiB4. Values of Posterior in parenthesis indicates posteriori errors. Detailed error reduction is described in Appendix B.

  (Modified) *(lines 450-452 in the new version):*
  Table 3. Original (Org) and optimized (Post) state vectors for Hyytiälä and Harvard Forest in different phenological stages as defined by SiB4. Values of Posterior in parenthesis indicate posterior errors. The definition of the prior values is outlined in Appendix A, and the error reduction is described in Appendix B.

- Fig 7a comment: the red and orange lines don't seem that different here, perhaps cite some calculated statistical significance to emphasize that they are different?

→ Indeed, the red and the orange lines at Hyytiälä are almost the same because optimized alphas (1316 for Growth, 1331 for Maturity phase) are similar to the original value (1400). Thus, we included $g_i$ statistics only for Harvard Forest in the text.

- In the manuscript:
(Modified) *(lines 465-467 in the new version):*
In contrast, when the optimized value of $\alpha$ is included (red line), the amplitude of $g_i$ is improved. Compared to the optimization that excluded $\alpha$, the MBE is reduced from 0.006 to 0.003 $mol\ m^{-2}\ s^{-1}$.

- Fig 7b: why not also show the equivalent to the orange lines for Harvard Forest? (i.e. with optimized f(Tcan) but original alpha.

→ We drew the orange line in Fig. 7(a) to check the improvement of $g_i$. However, $F_{COS}$ is influenced by many more factors (e.g. $g_s$ and $g_i$), and the effect of $\alpha$ cannot be isolated. For this reason, the orange line is missing in Fig. 7(b).

- Line 475: your result seems to imply that above-canopy RH is a better observational quantity to use to derive gs, but this is counterintuitive in that the 'gs' specifically involves resistance (or conductance) at the leaf surface, and so theoretically we should use RH at the leaf surface. One alternate explanation here is that it could be incorrect leaf temperature which can lead to a bias in leaf surface RH which propagates to gs.

→ Thank you for your suggestion. There are three reasons that could lead to $F_{LH}$ being overestimated: (1) water vapor flux in the boundary layer to the leaf surface is too large, (2) boundary conductance is too small, and (3) leaf surface temperature is too small. Although we cannot compare the observed leaf surface temperature with the estimated one due to the absence of observation, the temperature in the canopy air space fits well with the observed air temperature (See Figure 1 below). We speculate that the overestimated water vapor flux is the main reason because the observed RH above the canopy explains the diurnal fluctuation of $g_s$, as shown in Figure 10. As you mentioned, we theoretically should use the leaf surface RH. Since we could not obtain the observed leaf surface RH, we used the RH above the canopy instead. Furthermore, factor (2) about boundary conductance should be evaluated with observations.

- In the manuscript:
(Original) *(lines 475-478 in the old version):*

However, SiB4 still tends to overestimate gs in the morning and late afternoon. In contrast, when we base the gs calculation on the observed RH above the canopy, the diurnal cycle is better simulated (orange dashed line). This implies that SiB4 has the tendency to underestimate the humidity stress in the late afternoon when converting observed specific humidity above the canopy to humidity at leaf surface level.

(Modified) *(lines 503-511 in the new version):*
However, SiB4 still tends to overestimate $g_s$ in the morning and late afternoon. The overestimated $F_{LH}$ in SiB4 can result from three factors: (1) an overestimated water vapor flux in the boundary layer to the leaf surface, (2) an underestimated boundary conductance or, (3) an underestimated leaf surface temperature. Since we do not have observations of the leaf surface temperature, we confirmed that the estimated canopy temperature has a tight 1 to 1 relation with the observed air temperature. We speculate that the main reason for the overestimated $F_{LH}$ is the uncertain water vapor flux. When we base the $g_s$ calculation on the observed RH above the canopy, the diurnal cycle is better simulated (orange dashed line in Fig.10). The overestimated water vapor pressure implies that SiB4 tends to underestimate the humidity stress in the late afternoon when converting observed specific humidity above the canopy to humidity at leaf surface level. We suggest evaluating the boundary conductance (point (2) above) with observations.

- Line 484: what are the alternatives to 'stomatal transpiration'?

→ As mentioned in Sect.2.2.2, observation-based $g_s$ was calculated by estimated transpiration from observed evapotranspiration with fixed fitting equation (FG approach). When evaporation is larger, evapotranspiration can be larger. When we checked the ratio of evaporation to evapotranspiration in SiB4 at Hyytiälä, we found that the ratios are larger in three months: April, September, and October. This might result in overestimated $g_s$ in the observations because we used the same fixed fitting equation for the entire year. Therefore, the alternatives to 'stomatal transpiration' indicates evaporation here.

- In the manuscript:
(Modified) *(lines 513-517 in the new version):*
The optimized model still underestimates $g_s$ at Hyytiälä in April, September, and October (Fig. 8b). This might indicate that we did not properly separate stomatal transpiration rates from the observed latent heat flux. The simulated mean ratios of evaporation to evapotranspiration in these three months are 66 %, 60 %, and 95 %, respectively, and these values are higher compared to the other months (43 to 53 %). Thus, we speculate that the observed evapotranspiration does not solely represent stomatal transpiration in these months due to larger evaporation rates, leading to overestimated $g_s$ in the observations.

- Line 493: clarify 'indicating humidity stress only shortly at midday'. Do you mean that the impact of humidity stress is short-lived or only important around midday?

> → It means that the duration of humidity stress is simulated too short by SIB4. We modified the text:

- In the manuscript:
  (Original) *(lines 492-493 in the old version):*
  However, $g_s$ values are generally overestimated and SiB4 simulates two peaks during daytime, indicating humidity stress only shortly at mid-day.

  (Modified) *(lines 526-528 in the new version):*
  However, $g_s$ values are generally overestimated and SiB4 simulates two peaks during daytime. This indicates that humidity stress is only briefly occurring at mid-day in SiB4.

- Line 498: which 'pseudo-observations'? maybe just use 'observationally-derived X' where X is the quantity you're referring to here and mention it explicitly.
  > → Both have the same meaning. We changed the word 'pseudo-observations' to 'observation-based'.

- Line 515: and also consistent with Hu et al (2021, PNAS)

> → We added the new reference 'Hu et al., (2021)' in the manuscript.

- In the manuscript:
  (Original) *(lines 514-516 in the old version):*
  The higher uptake at high latitudes and lower uptake at the tropics are nevertheless consistent with inverse modelling results presented in Ma et al. (2021) and would help towards closing the COS budget.
  (Modified) *(lines 548-549 in the new version):*
  The higher uptake at high latitudes and lower uptake at the tropics are nevertheless consistent with inverse modelling results presented in previous studies (Ma et al., 2021; Hu et al., 2021) and would help towards closing the COS budget.

- Line 542: how does the improvement in b0 compare to night-time conductance values calculated for CLM by Lombardozzi et al 2017? (maybe this citation could be discussed earlier where you mention b0 results)

> → Thank you for your idea to compare with the previous study by Lombardozzi et al., 2017. They show a minimum stomatal conductance in a boreal needle-leaf evergreen forest of 0.008 mol m$^{-2}$ s$^{-1}$ with one observation and in a temperate broadleaf deciduous forest as 0.073 (mean) ± 0.084

(standard deviation) mol m$^{-2}$ s$^{-1}$ with 22 observations. Comparing our b$_0$ with the value from Lombardozzi et al. is problematic due to the limited number of observations (n=1). The results over a temperate broadleaf deciduous forest have large variations, which might vary with season, humidity, etc. Therefore, it is challenging to compare our results to their observations. We need seasonal and diurnal observations to evaluate our optimized $b_0$ results.

- Appendix A comment: I think your prior errors are based on the 'initial value +/- 1.5 state errors'? So for example prior alpha should then be 1400 +/- 700 (as is shown in Fig B1A for HYYT). But this is inconsistent with Table 3 where you list 1400 +/- 1000. Can you please clarify?

→ In Figure A1(a), the red line indicates the 75 percentile ranges from 400 to 2500. Therefore, we decide to use a prior alpha of 1400 +/- 1000.

**Technical Corrections**

- Line 109: replace 'heterogenic' with 'heterogeneous'. I think the former is more related to genetic/species aspects.

→ We replaced 'heterogenic' with 'heterogeneous'.

- Line 234: delete 'the', or add the word site after 'Harvard Forest'

→ We deleted articles in front of 'Harvard Forest'

- Line 264: delete 'to use'

→ As you suggested, we deleted it.

- Line 266: delete 'of' before 'uncertainties in GPP'

→ As you suggested, we deleted it.

- Line 338: change 'humidity impact' to 'humidity stress impact' GPP'

→ As you suggested, we changed the word.

- Line 340: Capitalize 'we'
    → We added the part of the sentence 'to account for the better humidity impact for the global COS leaf uptake'.
- In the manuscript:
    (Original) *(The line 340 in the old version):*
    we simulated the global COS leaf uptake without the 0.7 threshold of F$_{LH}$ for ENF.

    (Modified) *(lines 359-360 in the new version):*
    To account for the optimized humidity impact on the global COS leaf uptake,

we simulated the global COS leaf uptake without the 0.7 threshold of $F_{LH}$ for ENF.

- Fig 5 caption: replace 'in' with 'at'

→ We replaced every 'in' in front of 'Harvard Forest' with 'at' in the manuscript.

- Line 403: replace 'in' with 'at', and 'In' with 'At' (and in any other instances when mentioning the sites, such as lines 407, 413, 433, Fig 7 caption, 463...)

→ We replaced every 'in' in front of 'Harvard Forest' with 'at' in the manuscript.

- Line 411: 'higher and smaller respectively'

→ We added 'respectively' in the sentence.

- Line 430: replace 'pseudo-observations' with 'derived' or 'observationally-derived'

→ We replaced all 'pseudo-observations' with 'observation-based' in the manuscript.

- Line 491: 'At the Harvard Forest site..'

→ We deleted articles in front of 'Harvard Forest'

- Fig A1 caption: 'where the cost is minimized' or 'where the cost reaches a minimum value'.

→ We modified the part of the caption as 'where the cost is minimized'.

- In the manuscript:
(Original) *(line 570-574 in the old version):*
Figure A1: Cost function values plotted against the value of the state vectors elements in Hyytiälä (solid line) and Harvard Forest (dotted line). The red lines indicate a criteria cost calculated by *H(x)* as the 75-percentile value of every three-hourly observation in each month. While the target parameter changes, the other variables are fixed as $\alpha$ = 1400 (Hyytiälä), 2000 (Harvard Forest), $\Delta H_a$ = 40 kJ mol$^{-1}$ $\Delta H_{eq}$ = 100 kJ mol$^{-1}$, and $T_{eq}$ = 295 K (Hyytiälä), 310 K (Harvard Forest), $b_0$ = 0.02 (Hyytiälä), 0.01 (Harvard Forest), and $b_1$ = 17 (Hyytiälä), 12 (Harvard Forest). These values were decided where the cost has minimum.

(Modified) *(lines 606-610 in the new version):*
Figure A1: Cost function values plotted against the value of the state vector elements at Hyytiälä (solid line) and Harvard Forest (dotted line). The red lines

indicate a criteria cost calculated by *H(x)* as the 75-percentile value of every three-hourly observation in each month. While the target parameter changes, the other variables are fixed as $\alpha$ = 1400 (Hyytiälä), 2000 (Harvard Forest), $\Delta H_a$ = 40 kJ mol$^{-1}$ $\Delta H_{eq}$ = 100 kJ mol$^{-1}$, and $T_{eq}$ = 295 K (Hyytiälä), 310 K (Harvard Forest), $b_0$ = 0.02 (Hyytiälä), 0.01 (Harvard Forest), and $b_1$ = 17 (Hyytiälä), 12 (Harvard Forest). These values were based on the value where the cost reached a minimum.

- Line 585: Reiterating comments from above, GPP is not an observation.

→ Thank you for your remark. We rewrote the GPP part in the sentence.

- In the manuscript
  (Original) *(line 585-586 in the old version):*
  We optimized each ensemble with the same observations (GPP and COS leaf uptake) and state variables but added noise to each ensemble member (Chevallier et al., 2007).

  (Modified) *(lines 621-622 in the new version):*
  We optimized each ensemble with the same $y$ (observationally derived GPP and COS leaf uptake) and $x$ but added noise to each ensemble member (Chevallier et al., 2007).